# Assessment of the Weather Research and Forecasting (WRF) Model for Simulation of Extreme Rainfall Events in the Upper Ganga Basin

Ila Chawla[1,5], Krishna K Osuri[3,4], P P Mujumdar[1,2], and Dev Niyogi[4,5]

[1]Department of Civil Engineering, Indian Institute of Science, Bangalore, 560012, India
[2]Divecha Centre for Climate Change, Indian Institute of Science, Bangalore, 560012, India
[3]Department of Earth and Atmospheric Sciences, NIT Rourkela, Odisha, 769008, India
[4]Department of Agronomy- Crops, Soils, Water Sciences, Purdue University, West Lafayette, IN 47907, USA
[5]Department of Earth, Atmospheric and Planetary Sciences, Purdue University, West Lafayette, IN 47907, USA

*Correspondence to*: P P Mujumdar (pradeep@iisc.ac.in)

# Assessment of the Weather Research and Forecasting (WRF) Model for Simulation of Extreme Rainfall Events in the Upper Ganga Basin

Ila Chawla[1,5], Krishna K Osuri[3,4], P P Mujumdar[1,2], and Dev Niyogi[4,5]

[1]Department of Civil Engineering, Indian Institute of Science, Bangalore, 560012, India
[2]Divecha Centre for Climate Change, Indian Institute of Science, Bangalore, 560012, India
[3]Department of Earth and Atmospheric Sciences, NIT Rourkela, Odisha, 769008, India
[4]Department of Agronomy- Crops, Soils, Water Sciences, Purdue University, West Lafayette, IN 47907, USA
[5]Department of Earth, Atmospheric and Planetary Sciences, Purdue University, West Lafayette, IN 47907, USA

*Correspondence to*: P P Mujumdar (pradeep@iisc.ac.in)

**Abstract.** Reliable estimates of extreme rainfall events are necessary for an accurate prediction of floods. Most of the global rainfall products are available at a coarse resolution, rendering them less desirable for extreme rainfall analysis. Therefore, regional mesoscale models such as the advanced research version of the Weather Research and

Forecasting (WRF) model, are often used to provide rainfall estimates at fine grid spacing. Modeling heavy rainfall events is an enduring challenge, as such events depend on multiscale interactions, and the model configurations such as grid spacing, physical parameterization, and initialization. With this background, the WRF model is implemented in this study to investigate the impact of different processes on extreme rainfall simulation, by considering a representative event that occurred during 15 – 18 June 2013 over the Ganges basin in India, which is located at the

foothills of the Himalayas. This event is simulated with ensembles involving four different microphysics (MP), two cumulus (CU) parametrizations, two planetary boundary layer (PBL), and two land surface physics options; and different resolutions (grid spacing) within the WRF model. The simulated rainfall is evaluated against the observations from 18 rain gauges and the Tropical Rainfall Measuring Mission Multi-Satellite Precipitation Analysis (TMPA) 3B42 version 7 data. From the analysis, it is noted that the selection of MP scheme influences the spatial

pattern of rainfall, while the choice of PBL and CU parametrizations influence the magnitude of rainfall in the model simulations. Further, WRF run with Goddard MP, Mellor–Yamada–Janjic PBL and Betts–Miller–Janjic CU scheme is found to perform 'best' in simulating this heavy rain event. The selected configuration is evaluated for several heavy to extremely heavy rainfall events that occurred across different months of the monsoon season in the region. The model performance improved through incorporation of detailed land surface processes involving

prognostic soil moisture evolution in Noah scheme as compared to the simple Slab model. To analyze the effect of model grid spacing, two sets of downscaling ratios – (i) 1:3, Global to Regional (G2R) scale; and (ii) 1:9, Global to Convection-permitting scale (G2C) are employed. Results indicate that higher downscaling ratio (G2C) causes higher variability and consequently, large errors in the simulations. Therefore, G2R is opted as a suitable choice for simulating heavy rainfall event in the present case study. Further, the WRF simulated rainfall is found to exhibit less

bias when compared with the NCEP FiNaL (FNL) reanalysis data.

## 1. Introduction

Indian Summer Monsoon Rainfall (ISMR) is often associated with very heavy (124.5 to 244.4 mm/day) to extremely heavy (more than 244.5 mm/day) rainfall (Indian Meteorological Department, Terminologies and Glossary; http://imd.gov.in/section/nhac/termglossary.pdf), particularly during June to September months. The extremely heavy rainfall events usually occur due to the presence of organized meso-convective systems (MCSs) embedded in large-scale monsoonal features such as offshore troughs and vortices, depressions over the Bay of Bengal /Arabian Sea, and mid-tropospheric cyclones (Sikka and Gadgil, 1980; Webster et al., 1998; Fasullo and Webster, 2003).

Extremely heavy rainfall at shorter time scales are particularly difficult to predict in mountainous terrains, and continue to be a challenge to operational and research community (Das et al., 2008; Li et al., 2017). Global models have been employed in several studies to understand the large-scale circulation pattern and for quantitative analysis of the monsoon rainfall, but due to their coarse resolution, they are unable to represent the local to regional characteristics of monsoon rainfall. Regional models, on the other hand, can explicitly simulate the interactions between the large-scale weather phenomenon and regional topography, making the climate simulations reliable (Gadgil and Sajani, 1998; Ratna et al., 2011; Srinivas et al., 2013). Furthermore, regional models have a better representation of convection thus offsetting one of the major sources of errors and uncertainties in the global models. Therefore, regional models become a preferred choice to study seasonal monsoon rainfall.

The advanced research version of the Weather Research and Forecasting model (hereafter referred as the WRF model) is a regional popular community model that is widely used for both studying as well as forecasting a variety of high impact meteorological events, such as rainfall (Vaidya and Kulkarni, 2007; Deb et al., 2008; Kumar et al., 2008; Chang et al., 2009; Routray et al., 2010; Mohanty et al., 2012), tropical cyclones (Raju et al., 2011; Routray et al., 2016; Osuri et al., 2017b) and thunderstorms (Madala et al., 2014; Osuri et al., 2017a). Several works are reported in the literature which have considered the WRF model over the Himalayan region. Kumar et al., (2012) used the WRF model to simulate the cloudburst event of 2010 in the Leh area over the north-western Himalayan belt. While, Kumar et al., (2014) and Thayyen et al., (2013) used the WRF model to gain insight into the atmospheric processes and the MCSs that led to the 2010 Leh event. Similarly, Chevuturi et al., (2015) simulated the heavy precipitation event of September 2012 in the central Himalayas using the WRF model. Medina et al., (2010) used the WRF model to understand how topography and land surface conditions affect the extreme convection in western and eastern Himalayas. Particularly for the 2013 heavy rainfall episode in the Uttarakhand region, the WRF model is used in several studies, including those by Kotal et al., (2014); Vellore et al., (2016); and Hazra et al., (2017) to understand the physical processes leading to the event. Shekhar et al., (2015); Chevuturi and Dimri, (2016); and Dimri et al., (2016) performed in-depth synoptic analysis of the June 2013 heavy rainfall event using the WRF model. Rajesh et al., (2016) presented the role of land surface conditions in simulating the heavy rainfall event. Therefore, from the existing literature, it can be established that the regional model performs considerably well over the region. However, finding the optimal set of physics parameterization schemes (along with the selection of an appropriate model grid spacing/resolution) to simulate extreme/heavy rainfall events, and understanding the

effect of the combination of different parametrization schemes on rainfall estimates over the Indian monsoon region is still an active area of research.

Earlier studies such as by Krishnamurthy et al., (2009); Misenis and Zhang, (2010); Rauscher et al., (2010); Mohanty et al., (2012); and Chevuturi et al., (2015) indicated that heavy rainfall predictions can be improved through ensemble model techniques and fine grid resolution. However, the influence of the interaction between model parameterization schemes on mesoscale rainfall simulations over India is still an understudied issue. In particular, heavy rain simulations studies have reviewed the impact of individual parameterization options such as the Microphysics (MP) scheme (Rajeevan et al., 2010; Raju et al., 2011; Kumar et al., 2012), Cumulus (CU) parameterization scheme (Deb et al., 2008; Mukhopadhyay et al., 2010; Srinivas et al., 2013; Madala et al., 2014), Planetary Boundary Layer (PBL) scheme (Li and Pu, 2008; Hu et al., 2010; Hariprasad et al., 2014), and Land Surface Model (LSMs) options (Chang et al., 2009). However, the ensemble analysis that reviews the relative impact of different configurations, and the associated variability (uncertainty) is lacking. It is important to study the impact of different parameterizations in an ensemble mode because it is often likely that the performance of one scheme depends on other model configurations considered. For example, the conclusions regarding which CU scheme performs best would be intimately tied to the choice of the MP or land surface options considered in conducting the numerical experiments. With this perspective, this paper seeks to assess the sensitivity of the WRF model to predict heavy to extremely heavy rainfall episodes over the Ganges basin in the foothills of the Himalayas. Specific tasks undertaken in this work are: (i) quantitative verification of the WRF model to simulate an extremely heavy rainfall event; (ii) assessment of the sensitivity of the model simulated rainfall to different parameterization options, downscaling ratios, and land surface models; (iii) validate the selected configuration for other rainfall events over the region; and (iv) comparison of the WRF simulated rainfall with the global reanalysis data to investigate the impact of local versus global factors on rainfall simulations. A related objective is to provide suitable recommendations on a possible optimal choice for model configuration to simulate such heavy rainfall events in the region.

*Description of the heavy rainfall event*

The 2013 summer monsoon had a normal onset but the trough advanced rapidly, covering the whole of India by mid-June, instead of mid-July (Ray et al., 2014). This large-scale setting is thought to have created a platform for interaction of two synoptic scale events – northwest moving depression from the Bay of Bengal and preexisting westerly trough in mid-troposphere. Meteorological studies conducted over the region (Kotal et al., 2014; Ray et al., 2014; Chevuturi and Dimri, 2016; Rajesh et al., 2016) established that there was a monsoon low-pressure system during this period. The longitudinal time section for 850 hPa geopotential height along with anomaly averaged over $20° – 26°$ N showed high negative anomaly on 14 June, which migrated to the west, over $75°$ E by 17 June. The meridional wind anomaly within the belt of $35° – 45°$ N showed a westerly wave, moving from $10°$ E on 12 June to $70°$ E on 17 June. These two anomalies are found to be in phase, consequently causing interaction between the eastward moving trough in the mid-upper troposphere and westward-moving monsoon low in the lower troposphere. The monsoon low provided the moisture feed and the upper-level westerly trough provided the divergence to lift the

moisture. This whole system eventually led to an unanticipated heavy rainfall during 15 – 18 June 2013 in the
Kedarnath valley and adjoining areas in the state of Uttarakhand, India (Kotal et al., 2014; Ray et al., 2014; Chevuturi and Dimri, 2016; Rajesh et al., 2016). The region received rainfall greater than 370 mm in one day (17 – 18 June 2013), which is 375% above the daily normal rainfall (65.9 mm) during the monsoon season (Ray et al., 2014). Consequently, heavy floods occurred in the region, causing unprecedented damage to life and property.

The synthesis of the synoptic setting of the event has been carried out in a number of studies such as Dube et al.,
(2014); Kotal et al., (2014); Ray et al., (2014); Shekhar et al., (2015); Chevuturi and Dimri, (2016); and Rajesh et al., (2016), but the mesoscale assessment pertaining to the simulation of this rainfall event is still lacking. Therefore, the present study emphasizes on quantitatively evaluating and conducting the sensitivity analysis of the WRF model in predicting extreme rainfall. The ability of the WRF model to simulate heavy rainfall events is further verified by considering additional episodes (apart from the June 2013 event, details of which are presented in Section 2.1) that
occurred within this region across different monsoon months.

Study region comprising of the upstream part of the Ganga Basin in India, referred as Upper Ganga Basin (UGB) hereafter, is selected for the analysis in this paper. Figure 1 presents the topography of the UGB as described for the three domains of the WRF model (Domain 1, Domain 2a and Domain 2b with 27, 9 and 3 km grid resolution respectively) along with the 18 rain gauge stations located within the region. The region is of social, cultural and
economic importance to India, further making this study necessary.

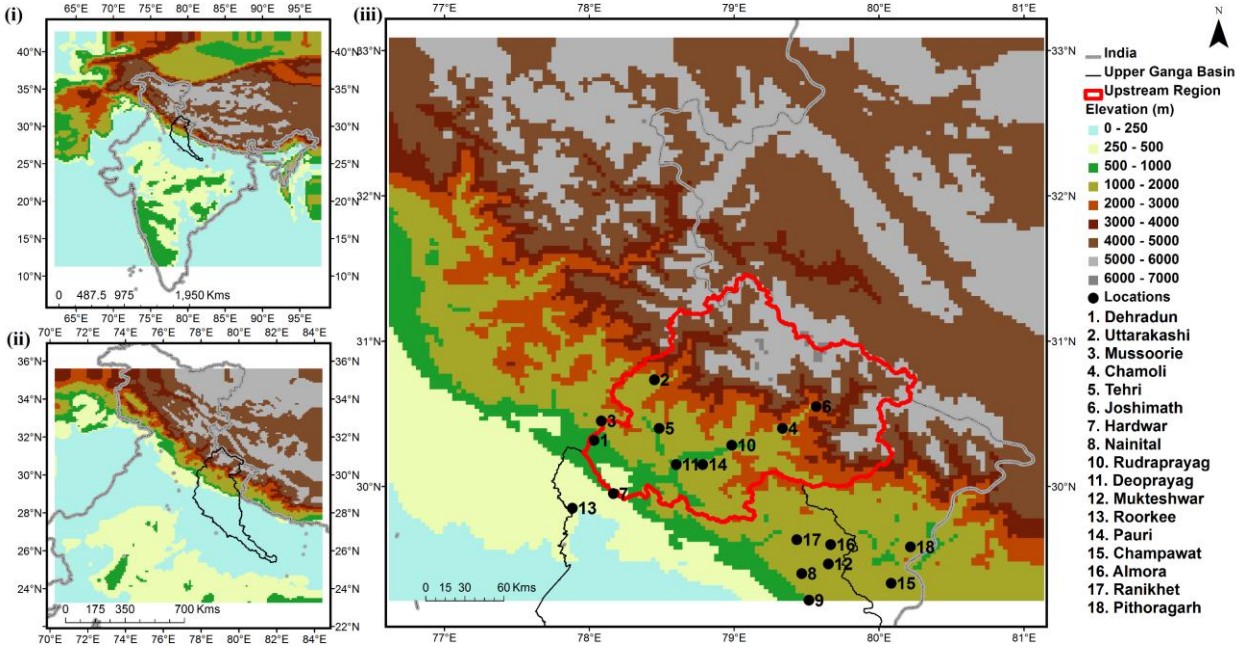

**Figure 1.** Topography of the study region (shown with black outline) as represented in the WRF model for (i) Domain 1 – 27 km grid spacing; (ii) Domain 2a – 9 km grid spacing (downscaling ratio – 1:3); and (iii) Domain 2b – 3 km grid spacing (downscaling ratio – 1:9). Locations of the rain gauge stations within the UGB are presented as black dots in Figure 1 (iii).

## 2. Data and Experimental Setup

### 2.1 Observed Data

Figure 2 presents daily and cumulative rainfall data from 15 to 18 June 2013 (obtained from the Indian Meteorological Department (IMD) and the literature (Ray et al., 2014) for the 18 official rain gauges located within the UGB.

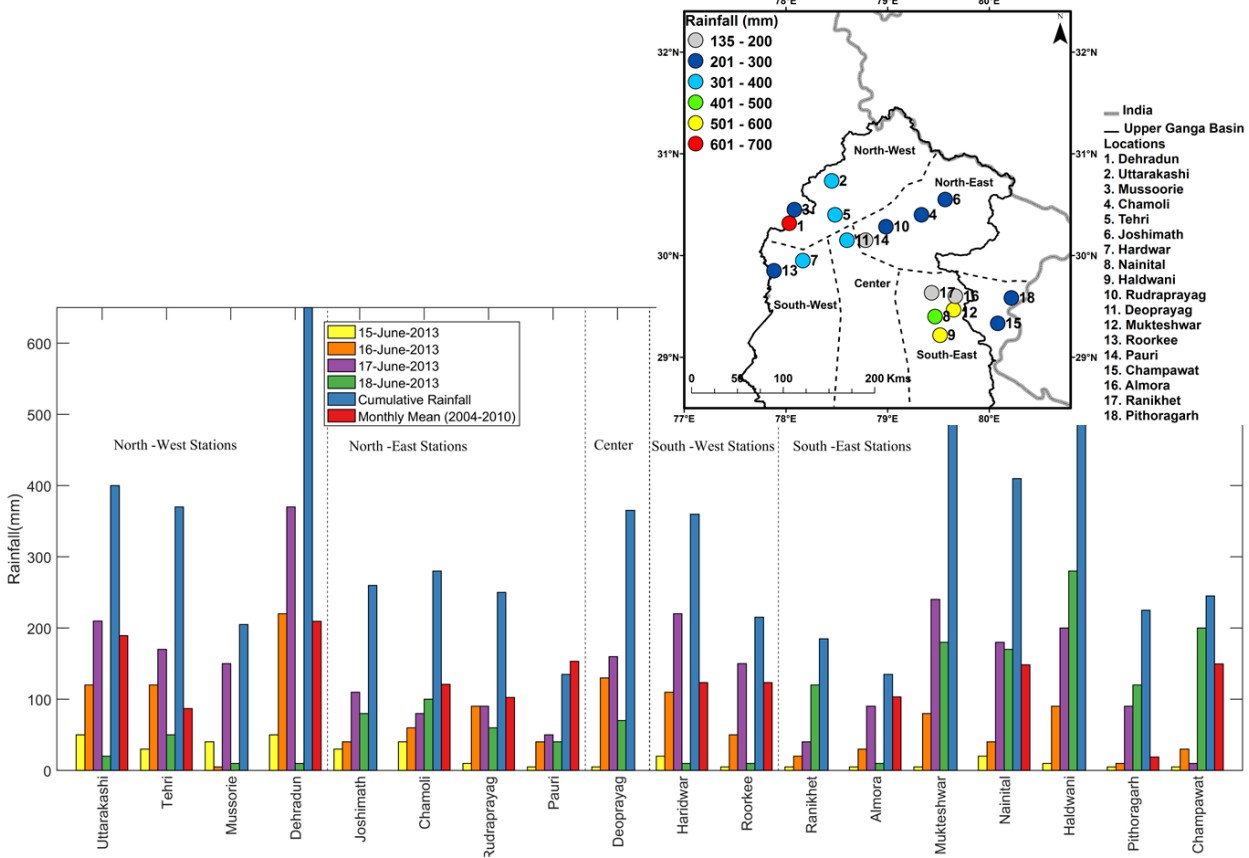

**Figure 2.** Observed daily and cumulative rainfall along with historic monthly mean (2004 – 2010) values at the 18 rain gauges in the upstream region of the UGB.

It is noticed that the northwest part of the region received higher rainfall compared to the northeast, with stations such as Tehri and Dehradun showing 327% and 210% (respectively) more rainfall than their historic means. A few stations like Chamoli in the northeast region received 250 mm of cumulative rainfall over the 4 days' period, which is 144% higher than the historic mean. In general, most of the stations in the southern part of the basin, which are located at a lower elevation, recorded relatively less rainfall with a cumulative range of 445 mm, in comparison to the northern part (at higher altitude) having a rainfall range of 515 mm. Additionally, three stations in the southeast region, i.e., Mukteshwar, Haldwani and Nainital received extremely heavy rainfall with a cumulative average of 498 mm. From the above analysis, it is evident that the system moved from east to west direction with two distinct regions in the UGB – southeast, and northwest, receiving extremely heavy rainfall.

The region has complex topography and a limited number of rain gauges because of the difficulty in operating a network in this region. To further capture the spatial variability in rainfall, Tropical Rainfall Measuring Mission

Multi-Satellite Precipitation Analysis (TMPA) 3B42 (version 7) product, which is available at 0.25° resolution at
daily scale is analyzed (Fig. 3). It is to be noted that, since the focus area for the analyses is the upstream region of
        the UGB (Fig. 1 (iii) and Fig. 2), results are presented with respect to the geographical extent of Domain 2b
        throughout this paper. From Figure 3, it can be noted that the TMPA data is able to capture the spatial variability in
        the rainfall – with distinct clusters corresponding to heavy rainfall in the northwest and southeast regions of the
        study area. However, the rainfall amount is significantly underestimated by the TMPA product, with the maximum
value of 265 mm against the recorded 650 mm. This under reporting for gridded satellite product versus rain gauge
        in the ISMR region is a well-known feature (Rahman et al., 2009; Mishra and Srinivasan, 2013; Kneis et al., 2014;
        Bharti et al., 2016). The TMPA estimates are verified against the IMD station observations for baseline quality
        check. Mean absolute error (*MAE*), root mean square error (*RMSE*) and bias (*β*) are computed using the nearest
        neighbourhood mapping approach and are presented in Table 1.

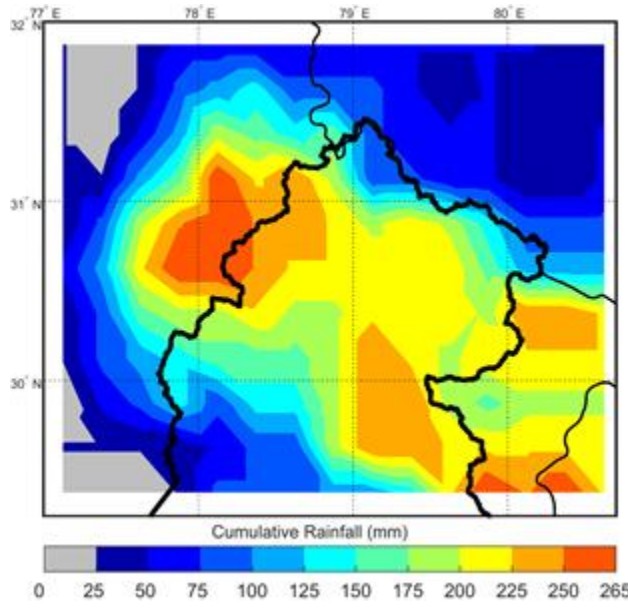

**Figure 3.** Cumulative rainfall in the upstream region of the UGB obtained from the Tropical Rain Measurement Mission
(TMPA) 3B42V7 product at 0.25° resolution. The domain shown is similar to the insert shown in Figure 2.

**Table 1.** Comparison of TMPA data with station data

| Station | Mean Absolute Error (mm) | Root Mean Square Error (mm) | Bias* (%) |
|---|---|---|---|
| Uttarakashi | 64 | 97 | -40 |
| Tehri | 63 | 83 | -44 |
| Mussorie | 75 | 94 | -6 |
| Dehradun | 126 | 191 | -70 |
| Joshimath | 51 | 55 | -21 |
| Chamoli | 44 | 55 | -27 |
| Rudraprayag | 48 | 53 | -22 |
| Pauri | 34 | 44 | 37 |
| Deoprayag | 95 | 98 | -56 |
| Hardwar | 83 | 114 | -76 |
| Roorkee | 64 | 82 | -39 |

| | | | |
|---|---|---|---|
| Ranikhet | 71 | 77 | 27 |
| Almora | 26 | 30 | 63 |
| Mukteshwar | 90 | 118 | -60 |
| Nainital | 86 | 111 | -67 |
| Haldwani | 115 | 157 | -80 |
| Pithoragarh | 56 | 69 | -6 |
| Champawat | 100 | 126 | 1 |

$*Bias\ (\%) = \frac{(Station\ data - TRMM\ data)}{Station\ data} \times 100$

TMPA data is observed to behave differently for different ranges of rainfall values over the study domain. TMPA overestimated the rainfall at stations with cumulative rainfall less than 200 mm (e.g., Pauri and Almora). In contrast, rainfall at stations receiving more than 250 mm of cumulative rainfall is underestimated. Stations that received rainfall of 200 – 250 mm are well represented in the TMPA data (e.g., Mussorie, Pithoragarh, and Champawat). From the analysis, it could be inferred that in the TMPA data rainfall values are clustered towards the mean value.

Errors noticed in the TMPA data could be attributed to two factors: first, large spatial coverage and coarse resolution of the TMPA data, and second, for comparison with the observed data a simple approach of selecting nearest grid point is implemented.

In addition to June 2013 case, five additional heavy to extremely heavy rainfall events are also considered in the present study for the analysis, details of which are presented in Table 2. Rainfall from the IMD gridded data at 0.25° 
resolution (Pai et al., 2014) is considered as the observed data for these events.

**Table 2.** Heavy to extremely heavy rainfall events recorded in the UGB region

| Event No. | Time Period | Maximum Rainfall Day | Maximum Rainfall Amount (mm) |
|---|---|---|---|
| 1 | 18 – 22 June 2008 | 20 June | 126 |
| 2 | 29 July – 2 August 2010 | 31 July | 271 |
| 3 | 15 – 19 August 2011 | 16 August | 234 |
| 4 | 17 – 21 September 2010 | 19 September | 218 |
| 5 | 11 – 15 September 2012 | 14 September | 38 |

It is to be noted that on 13 – 14 September 2012, cloudburst event was reported in the region and the total amount of rainfall on 14 September was recorded approximately to be 210 mm (Chevuturi et al., 2015). This event is 
significantly underestimated in the IMD gridded data, indicating that caution must be exercised while using the data for applications involving heavy rainfall events, such as flood modeling and validating the rainfall simulations from the mesoscale models. Figure 4 presents the spatially averaged daily and cumulative rainfall received during different events (as specified in Table 2).

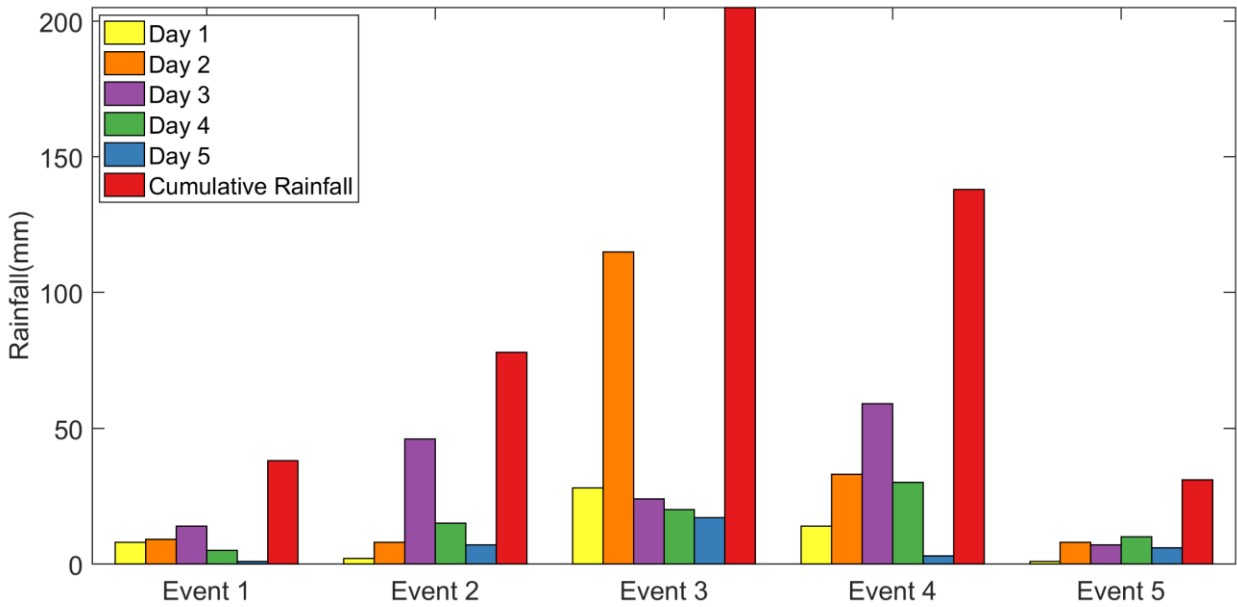

**Figure 4.** Spatially averaged daily and cumulative rainfall for Event 1 (18 – 22 June 2008); Event 2 (29 July – 2 August 2010); Event 3 (15 – 19 August 2011); Event 4 (17 – 21 September 2010); and Event 5 (11 – 15 September 2012) in the upstream region of the UGB.

## 2.2 Model Configuration and Experimental Setup

The simulation experiments in this paper are conducted using the WRF model, version 3.8. WRF is a Numerical Weather Prediction (NWP) non-hydrostatic, mesoscale model, available with several advanced physics and numerical schemes, designed for better prediction of atmospheric processes. The model description and updates can be found from Skamarock et al., (2005) and the WRF user webpage (http://www2.mmm.ucar.edu/wrf/users/).

The WRF model utilizes large-scale atmospheric forcing as input for initialization and lateral boundary condition. These large-scale conditions are regridded by the model domain considering the grid spacing, and local topographical as well as other terrain conditions. As is common for most WRF studies over the Indian region, National Centers for Environmental Prediction (NCEP) global FiNaL (FNL) analysis dataset, based on Global Data Assimilation System (GDAS) with Global Forecast System (GFS) is considered. The FNL data is available at a coarse resolution of $1° \times 1°$, at every six hours' interval – 00, 06, 12 and 18 UTC (Coordinated Universal Time) and is used to provide initial and boundary conditions to the model. The lateral boundary conditions in the WRF model are updated at 6-h intervals. Considering the short duration of the run, the model was forced with fixed Sea Surface Temperature (SST) throughout the integration, and no regional data assimilation is carried out. The land surface boundary conditions are taken from the Moderate Resolution Imaging Spectroradiometer (MODIS) International Geosphere–Biosphere Programme (IGBP) 21-category landuse/cover fields that are available with a horizontal grid spacing of 10 min. Three telescopically-nested domains are used in this study – the parent domain (Domain 1) is fixed between 60°E and 100°E with grid-spacing of 27 km; the first nested domain (Domain 2a) covers 70-85°E, 22-37°N with 9 km grid spacing and is indicative of "global to regional scale" (G2R) downscaling; and the second nested domain (Domain 2b) covers 76-81.5°E and 28.5-34°N at 3 km grid spacing (Fig. 1), for "global to convective scale" (G2C) downscaling (Trapp et al., 2007). The parent domain provides lateral boundary conditions to the inner

domains, resulting in the downscaling ratios for simulations as 1:3 and 1:9. The three domains use 30 vertical
pressure levels, with the top fixed at 50 hPa. The model time steps were a function of grid spacing: 135 s, 45 s and
15 s respectively for the three domains.

The model configuration used default parameterization options following Osuri et al., (2012). For example,
shortwave radiation is based on Dudhia (Dudhia, 1989), and longwave based on Rapid Radiative Transfer Model
(RRTM; (Mlawer et al., 1997)) scheme. Other physical parameterization options such as Microphysics (MP),
Cumulus (CU) parameterization schemes, Planetary Boundary Layer (PBL) and Land Surface Models (LSMs) were
selected as outlined ahead. There is currently no known unique configuration that can best simulate an extremely
heavy rainfall event. Therefore, based on the literature (e.g. (Kumar et al., 2008; Hong and Lee, 2009; Misenis and
Zhang, 2010; Mukhopadhyay et al., 2010; Argüeso et al., 2011; Cardoso et al., 2013; Efstathiou et al., 2013)), four
MP schemes, two CU schemes, two PBL schemes and two LSMs are considered to obtain an ensemble of rainfall
simulations. The two PBL schemes considered are the Yonsei University (YSU) scheme (Hong et al., 2006) and the
Mellor–Yamada–Janjic (MYJ) scheme (Janić, 2001). YSU is a non-local scheme, wherein fluxes are calculated at a
certain height in the PBL considering the profile of the entire domain. MYJ scheme, on the other hand, is a local
scheme in which fluxes are calculated at various heights within the PBL and are related to vertical gradient in the
atmospheric variables at the same height. Further details regarding the difference between the YSU and the MYJ
schemes can be obtained from Misenis and Zhang, (2010); and Efstathiou et al., (2013). The two CU schemes
considered are the Kain – Fritsch (KF) scheme (Kain, 2004) and the Betts-Miller-Janjic (BMJ) scheme (Janjić, 1994,
2000). The KF scheme is both shallow and deep convection scheme, wherein shallow convection is allowed for
updrafts that do not reach minimum precipitating cloud depth. This scheme is based on entrainment and detrainment
plume model with updrafts and downdrafts of mass flux. Potential energy is removed in the convective time scale
within this scheme. Furthermore, it includes cloud, rain, snow and ice detrainment at cloud top. BMJ, on the other
hand, considers convection at both shallow and deep levels. However, there is no updraft and downdraft of mass
flux and no cloud detrainment. Domain 2b is configured without any CU scheme, assuming MP to explicitly solve
the convection at the finer resolution (Sikder and Hossain, 2016). Four MP schemes considered are, the Purdue Lin
(PLin) scheme (Lin et al., 1983; Chen and Sun, 2002), the Eta Ferrier (Eta) scheme (NOAA, 2001), the WRF
Single-Moment 6-class (WSM6) scheme (Hong and Lim, 2006) and the Goddard scheme (Tao et al., 1989). Both,
the PLin scheme and the WSM 6 scheme are based on the parameterization from Rutledge and Hobbs, (1984) and
have 6-class microphysics, which includes the mixing ratios of water vapor, cloud water, cloud ice, snow, rain, and
graupel. The main difference between these two schemes is related to the treatment of ice-phase microphysical
processes. Details of the PLin and the WSM6 schemes are available in (Hong et al., 2009). The Eta scheme was
designed primarily for computational efficiency in NWP models, wherein the total condensate and the water vapors
are directly advected into the model. The Goddard scheme is a slight modification from the PLin scheme for ice-
water saturation. In general, all the MP schemes are known to influence the rainfall simulations at fine grid
resolution by influencing the water phase component (Li et al., 2017). Since each physics scheme is associated with
a distinct feature, it is important to examine the effect of their interactions on the rainfall simulations.

The sensitivity of various WRF configurations to simulate heavy rainfall events is assessed using the Noah LSM (Chen and Dudhia, 2001; Ek et al., 2003; Tewari et al., 2004). The Noah LSM is a community model that is included in the WRF suite with the prime aim of providing reliable boundary conditions to the atmospheric model. As a result, Noah LSM is a moderately detailed model, which includes single canopy layer with canopy resistance scheme of Noilhan and Planton, (1989) and four soil layers (at 0.1, 0.3, 0.6 and 1.0 m) with a total soil depth of 2 m. The last soil layer of 1 m acts as a reservoir for drainage of water under gravity and the above three layers serve as root zone depths. There is a provision in the model to change default root zone depths with the actual values from the field, subjected to data availability. In the Noah LSM, surface (skin) temperature is obtained using a single linearized surface energy balance equation, which effectively considers the ground and vegetation surface. Frozen soil parametrization based on Koren et al., (1999) and surface runoff scheme of Schaake et al., (1996) are also included in this model. Soil moisture, soil temperature, water intercepted by the canopy and snow stored on the ground are also included as the prognostic variables in the model. More detailed information on the Noah LSM can be obtained from Ek et al., (2003).

To assess the effect of the land surface scheme on simulations, the Noah LSM is replaced with the simple five-layer Soil Model (Slab; (Dudhia, 1996)). In contrast to the relatively sophisticated Noah LSM, Slab is based on simple thermal diffusion in the soil layers that has constant soil moisture availability but a prognostic soil temperature term (Deardorff, 1978). Further differences between the two LSMs are presented in Section 3.1.3.

Table 3 provides the summary of the WRF physics schemes considered to simulate the extremely heavy rainfall events.

**Table 3.** Configuration of the WRF model considered for simulation of rainfall

| Model Options | Dataset/Value |
|---|---|
| Domains | 3 |
| Grid Resolution (spacing) | 27 km; 9 km; 3 km |
| Downscaling ratio | 1:3; 1:9 |
| Projection System | Mercator |
| Land Surface Boundary Condition | 21-class MODIS |
| Initial Conditions | NCEP FNL |
| Short Wave Radiation Scheme | MM5 Shortwave or Dudhia |
| Long Wave Radiation Scheme | Rapid Radiative Transfer Model (RRTM) |
| PBL Schemes | 1. Yonsei University (YSU) <br> 2. Mellor–Yamada–Janjic (MYJ) |
| Cumulus Schemes | 1. Kain-Fritsch (KF) <br> 2. Betts-Miller-Janjic (BMJ) |
| Microphysics Schemes | 1. Lin (Purdue) <br> 2. Eta (Ferrier) <br> 3. WSM 6 <br> 4. Goddard |
| Surface Layer Option | Monin-Obukhov Similarity Theory |
| Land Surface Models | 1. Simple 5-layer Soil Model (Slab) <br> 2. Noah |

Ability of the WRF model configuration to simulate an extreme rainfall event is evaluated by comparing the simulated rainfall with the observations through indices such as Scale Error (*SE*), which is the ratio of standard

deviation of model simulations to the observed standard deviation and Coefficient of Variation (*CV*) in addition to *MAE*, *RMSE* and *β*.

## 3. Results and Discussion

280

### 3.1 Sensitivity Analysis

### 3.1.1 Verification of WRF Simulations

Figure 5 presents cumulative rainfall for 15 – 18 June 2013 from 16 WRF simulations (4 MP, 2 CU and 2 PBL) corresponding to each of the three domains.

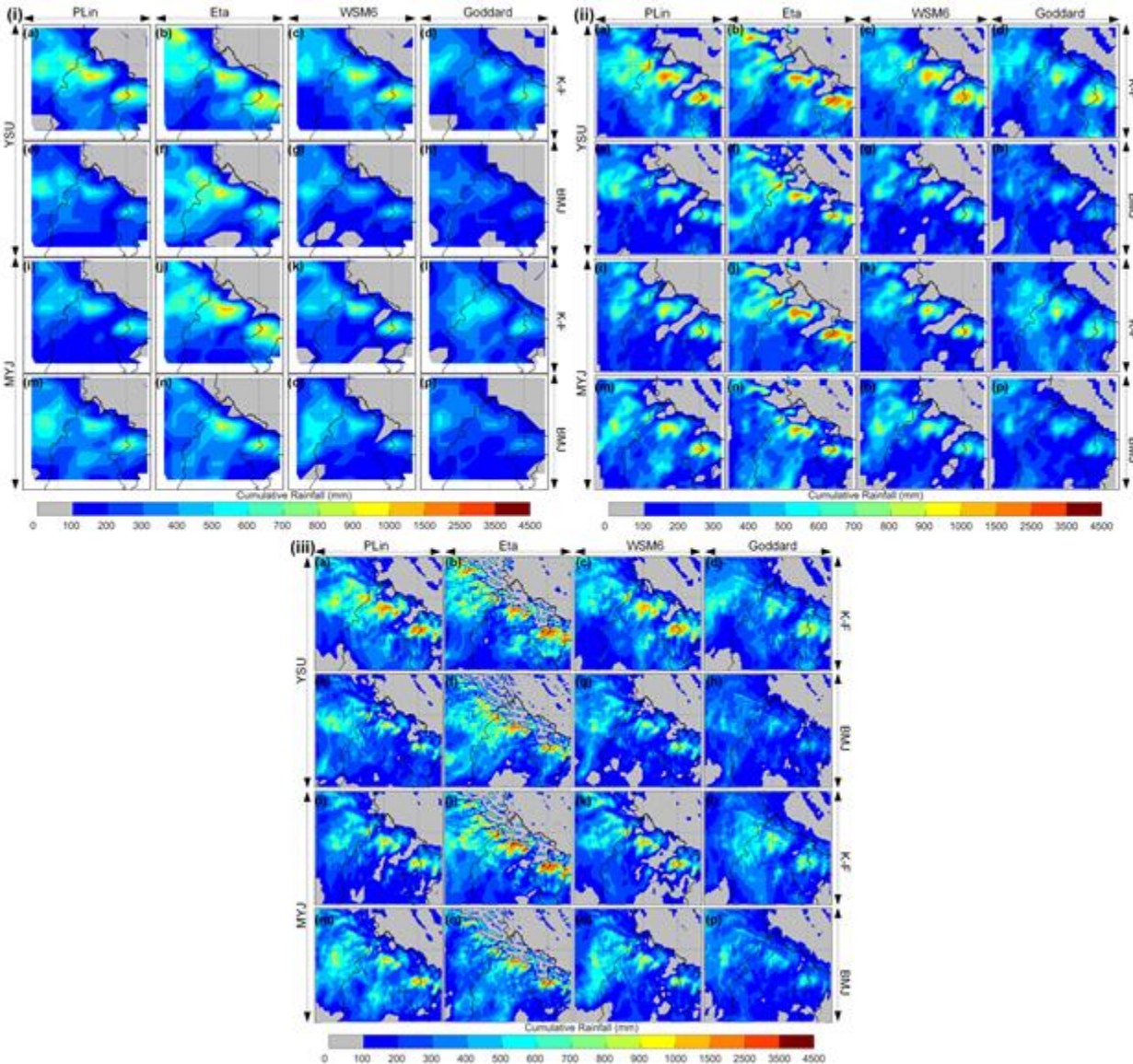

285

**Figure 5.** Spatial plots showing rainfall estimates obtained for (i) Domain 1; (ii) Domain 2a; and (iii) Domain 2b. Arrows in the left indicate the PBL scheme, arrows in the right represent the CU scheme and the top arrows present the MP scheme considered for the simulation runs. (a) to (p)* are the WRF configurations, for instance, Figure 5 (i) – (a) represents the WRF configuration with YSU PBL scheme, KF CU scheme and PLin MP scheme for Domain 1.

 *Refer to Appendix A (Table A.1) for the list of the WRF configurations.

From Figure 5 (i) to (iii), it may be seen that the spatial pattern of rainfall appears to be sensitive to the microphysics, i.e., PLin, Eta, and WSM6 MP schemes, while the amount of rainfall is more dependent on the PBL and CU scheme options. There is a considerable difference in the rainfall amount simulated with the Goddard MP scheme option as compared to other MP schemes. Further, most of the model runs are able to reproduce the spatial gradient in the rainfall amount, which is perhaps primarily due to the topographical variation in the region. For locales below 1000 m, observations show distinctly lower rainfall as compared to the high elevation regions (> 1000 m). Further, distinct clusters corresponding to heavy rainfall event are observed in the northeast and northwest areas of the study region. These clusters are found to be consistent with the TMPA data; however, due to lack of surface rain gauge observations, amount of rainfall in these regions could not be verified at this stage. Incidentally, the observed heavy rainfall event in the southeast part of the region is seen in a few WRF configurations, such as configuration (b) and (c). In general, WRF simulated rainfall fields show a similar spatial pattern as that of TMPA rainfall product. However, the magnitude of WRF rainfall is significantly high as compared to TMPA and is attributed to the negative bias in TMPA for heavy rains. Figure 6 summarizes the comparison of WRF rainfall with rain gauge observations, accumulated over the 4 days' period (15 – 18 June 2013) for the three domains. For comparison, grid points from the WRF domains closest to the gauge location are considered.

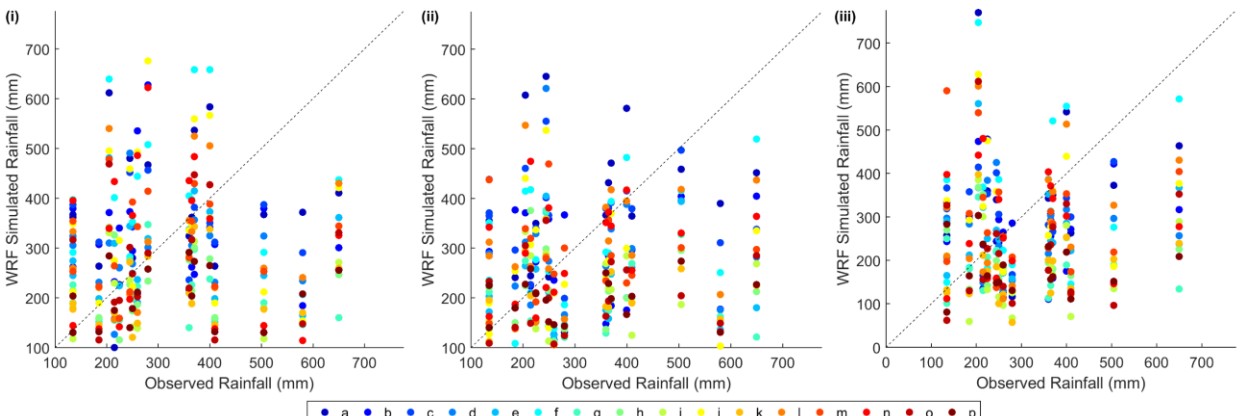

**Figure 6.** Scatter plots between the rainfall data from the rain gauges and the WRF simulations for (i) Domain 1; (ii) Domain 2a; and (iii) Domain 2b for (a) to (p)* WRF configurations.
*Refer to Appendix A (Table A.1) for the list of the WRF configurations.

Figure 6 indicates that Domain 1 captures rainfall within the range of 150 to 400 mm for most of the WRF configurations. For Domain 2a and Domain 2b, increase in the predicted rainfall amount is noted, particularly; for small rainfall thresholds. Further, the WRF runs still under predict extremely heavy rainfall and each of the configuration considered (across all the three domains) underestimated the rainfall amount more than 400 mm. However, the underestimation of rainfall is less in Domain 2b (G2C scale) compared to others, indicating the necessity of finer grid spacing as the first-order requirement for simulating the magnitudes of the extremely heavy rainfall events. The bias in the WRF simulations is typically due to number of interactive factors: (i) scale feedback between mesoscale convection and large-scale processes within the model (Bohra et al., 2006); (ii) lack of local observations that can add mesoscale features (Osuri et al., 2012; Osuri et al., 2015); (iii) lack of proper land surface

processes (Niyogi et al., 2006; Chang et al., 2009; Osuri et al., 2017a); and (iv) inability of the model to fully resolve

the complex topography (Argüeso et al., 2011; Cardoso et al., 2013; Chevuturi et al., 2015). To assess the

performance of the WRF simulations, quantitative scores (*MAE* and *RMSE*) with respect to the observed data are

computed for daily rainfall data, which is then averaged over the 4 days' period. The results are shown in Figure 7.

The last column in the figure presents the spatially averaged values obtained for different model configurations.

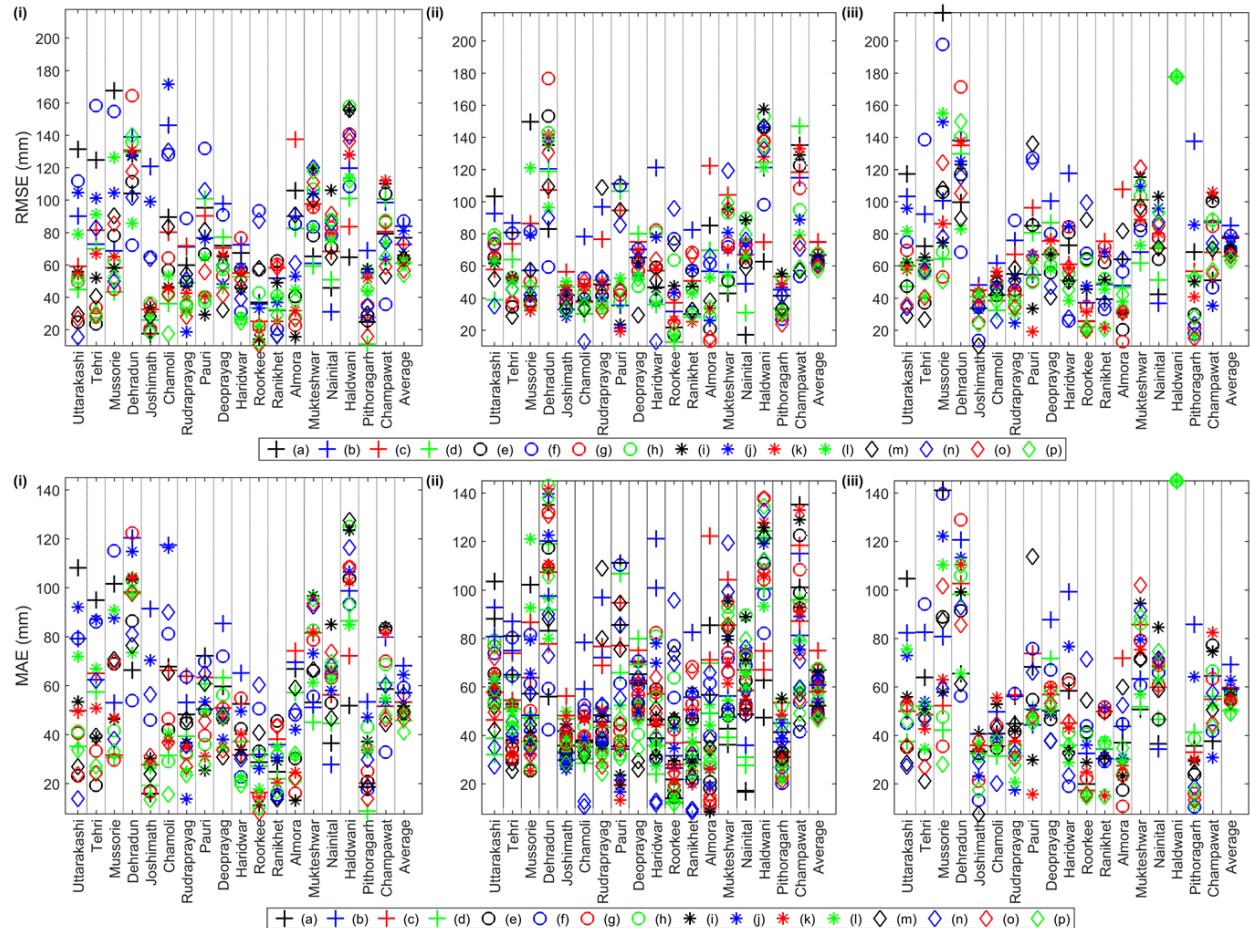


**Figure 7.** Root mean square error (top panel) and mean absolute error (bottom panel) computed temporally for (i) Domain 1; (ii) Domain 2a; and (iii) Domain 2b for (a) to (p)* WRF configurations.
*Refer to Appendix A (Table A.1) for the list of the WRF configurations.

Figure 7 indicates that there is more error at the stations Dehradun and Haldwani, which received higher rainfall.

The highest rainfall obtained in different WRF configurations for these stations was less than 500 mm and this

underestimation is highlighted in the error statistics. The model results show higher error and variability in the

simulations for the northern part of the domain as compared to southern. This is likely due to the complex terrain in

the northern part of the domain.

To identify the 'best' and the 'worst' performing configurations, temporal errors (wherein *MAE* across the 18

locations are summed up) and spatial errors (wherein *MAE* obtained over the entire region for the 4 days' is summed

up) for all the three domains are obtained (Appendix B), which indicate that the configuration (b) with YSU PBL,

KF CU, and Eta MP, produces maximum error, whereas configuration (p) with MYJ PBL, BMJ CU, and Goddard MP, gives minimum error.

To further assess the sensitivity of configuration (p) and configuration (b) in capturing the extreme rainfall events in
the region, additional simulations pertaining to other heavy to extremely heavy rainfall events (as mentioned in Table 2) are conducted. Spatial plots showing the cumulative rainfall estimates obtained for the three domains in comparison to the observed IMD gridded data and the TMPA data are presented in Appendix C. To summarize the performance of configuration (p) and configuration (b) against the observations (IMD gridded data), spatio-temporal MAE values are computed, which are presented in Table 4.

**Table 4.** *Spatio-temporal Mean Absolute Error (MAE) values (in mm) corresponding to WRF configuration (p) and configuration (b) for the three domains*

| Event No. | Domain 1 | | Domain 2a | | Domain 2b | |
|:---:|:---:|:---:|:---:|:---:|:---:|:---:|
| | *(p)* | *(b)* | *(p)* | *(b)* | *(p)* | *(b)* |
| 1 | 10 | 13 | 10 | 14 | 11 | 14 |
| 2 | 18 | 23 | 18 | 23 | 21 | 22 |
| 3 | 39 | 45 | 38 | 44 | 40 | 46 |
| 4 | 23 | 28 | 23 | 28 | 24 | 29 |
| 5 | 12 | 12 | 9 | 13 | 12 | 11 |

From the analysis conducted over the additional rainfall events, it is noted that configuration (p) gives less error in comparison to the configuration (b) for all the rainfall events. This makes configuration (p) with MYJ PBL, BMJ
CU, and Goddard MP the 'best' in simulating the spatial and temporal variability of the extremely heavy rainfall over the upstream region of the UGB. Why this combination emerged as the best performing is an intriguing but difficult question to address at this stage. Note that the rainfall prediction is the combination of many nonlinear, interactive factors including the behavior of each configuration and cannot be realistically studied with the sparse rainfall data and absence of vertical sounding observations. Some possible factors that could contribute would be
that local boundary formulation in MYJ may be more appropriately capturing the vertical environment in the complex terrain as compared to the nonlocal YSU scheme which seeks to simulate vertical mixing and boundary layer evolution using averaged and grid representative fields. As regards to the BMJ CU emerging in the top configuration, there are a number of studies for the ISMR where it has emerged as performing "overall best" (Vaidya and Singh, 2000; Ratnam and Kumar, 2005; Vaidya, 2006; Rao et al., 2007; Kumar et al., 2010;
Mukhopadhyay et al., 2010; Srinivas et al., 2013; Sikder and Hossain, 2016). As for the MP scheme, there are limited studies in comparison to those that have studied the CU configuration for the ISMR. Further, the MP scheme performance has been evaluated for tropical cyclone cases because of the warm versus cold pool processes that are critical in the simulation of the cyclone intensity. Of those available in the literature, studies such as Sing and Mandal, (2014) found that the Goddard scheme has a "slightly better" performance than other schemes. This
conclusion is also supported by studies such as Choudhury and Das, (2017) and has been used in hailstorm studies such as Chevuturi et al., (2014).

The impact of downscaling ratio on the rainfall simulations is addressed next. On comparing the simulations of June 2013 event from G2R and G2C domains with the rain gauge data, it is noted that the former gives less error for most of the locations (Appendix D). The G2C scale has large resolution (grid spacing) gap from outer to the inner domain

in comparison to G2R, which could result in less accurate initial and lateral boundary conditions, and consequently, more simulation errors in G2C. Another possibility is that the metric being used, which is the rainfall observation from in-situ data, itself is more conservative with regards to the grid in which rainfall occurs in the coarser domain and may slightly favor the G2R. However, on reviewing the overall structure of rainfall fields and the amounts across the domain, results suggest that the G2R scale with moderate downscaling ratio may be better suited for

simulation of the extreme rainfall event as in the present case study. The results are found to be consistent with other studies, such as by Liu et al., (2012), wherein the moderate ratio of 1:3 is found to perform best. However, it is to be noted that errors corresponding to the grid point nearest to the rain gauge are considered here for comparison. The result may vary upon selection of another grid point.

### 3.1.2 Impact of Different Parametrization Schemes

Although configuration (p), with MYJ PBL, BMJ CU, and Goddard MP, appears to be the 'best' physics configuration for the study region, significant variability exists among the simulations pertaining to different configurations of the WRF model. This variability causes significant uncertainty across different runs, which is quantified through computation of *SE and CV* (Fig. 8 and 9) for the June 2013 event. Deviation in model simulations with respect to observed data provides the *SE*, however, *CV* gives variation within different model simulations.

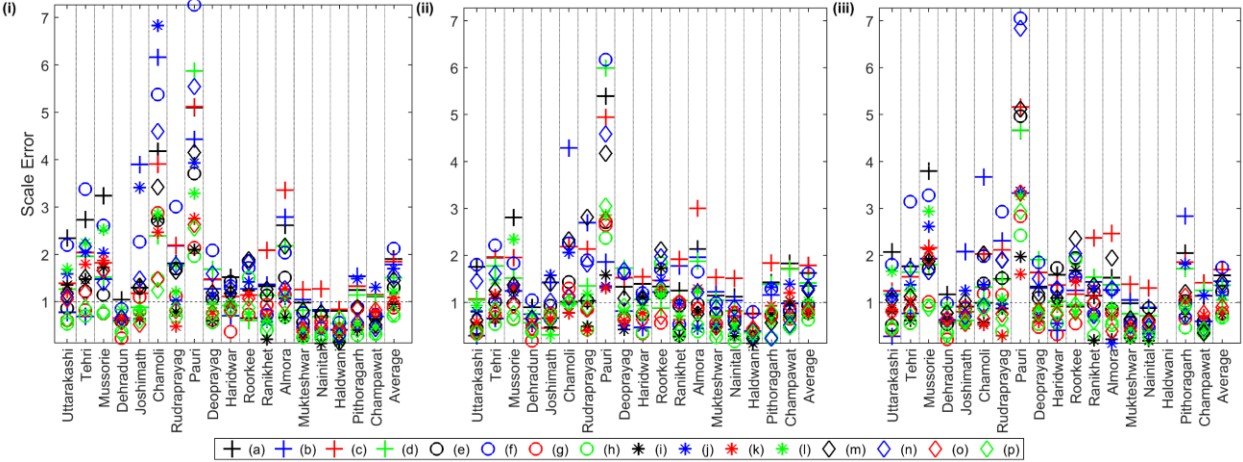


**Figure 8.** Scale error (*SE*) in (a) to (p)* WRF configurations for 18 locations in the UGB for (i) Domain 1; (ii) Domain 2a; and (iii) Domain 2b.

*Refer to Appendix A (Table A.1) for the list of the WRF configurations.

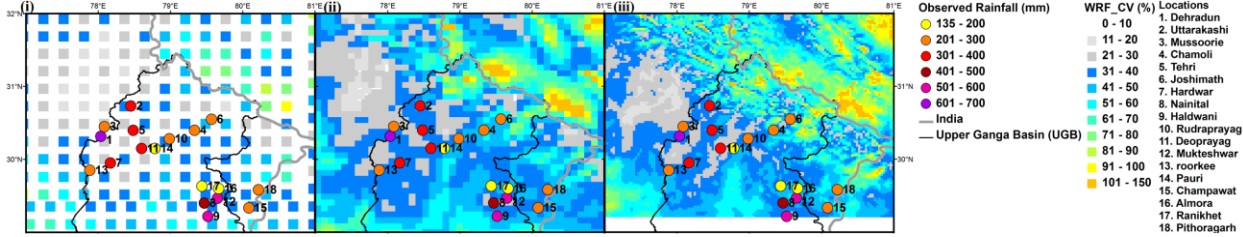

**Figure 9.** Coefficient of Variation (*CV*) value across different WRF configurations in the UGB for (i) Domain 1; (ii) Domain 2a; and (iii) Domain 2b.

Most of the model configurations have *SE* value clustered around 1 (Fig. 8), indicating that the variability in simulated rainfall is similar to the observed rainfall. However, variability in the northeastern part of the domain is observed to be high compared to others. Same is reflected in the *CV* plot (Fig. 9), wherein grid points around the Chamoli station (on the northeastern side) have *CV* between 41-51 %, whereas stations closer to Uttarakashi and Tehri have values ranging between 11 − 30 %. Further, grid points closer to Dehradun have low *CV* value, which could be due to the models consistently underestimating the rainfall in this subdomain. The southern part of the region, which received low rainfall, also exhibited high variability. In general, it can be inferred that uncertainty in rainfall is more in the northeastern part compared to the northwest. The regions that received very high or very low rainfall during this period also displayed higher uncertainty. Uncertainty in rainfall simulations varies between the domains, with Domain 2b having maximum uncertainty. This could be attributed to high variability in the simulated values at higher spatial resolution.

Since consideration of different parametrization schemes is the reason for variability in rainfall simulations, it is of interest to understand how former influences the model output. For this, the average cumulative rainfall over the region, across different configurations is considered. The differences between various configurations are evaluated to assess the influence of PBL, CU and MP parametrization schemes on the rainfall simulations. Results for the same are presented in Figure 10.

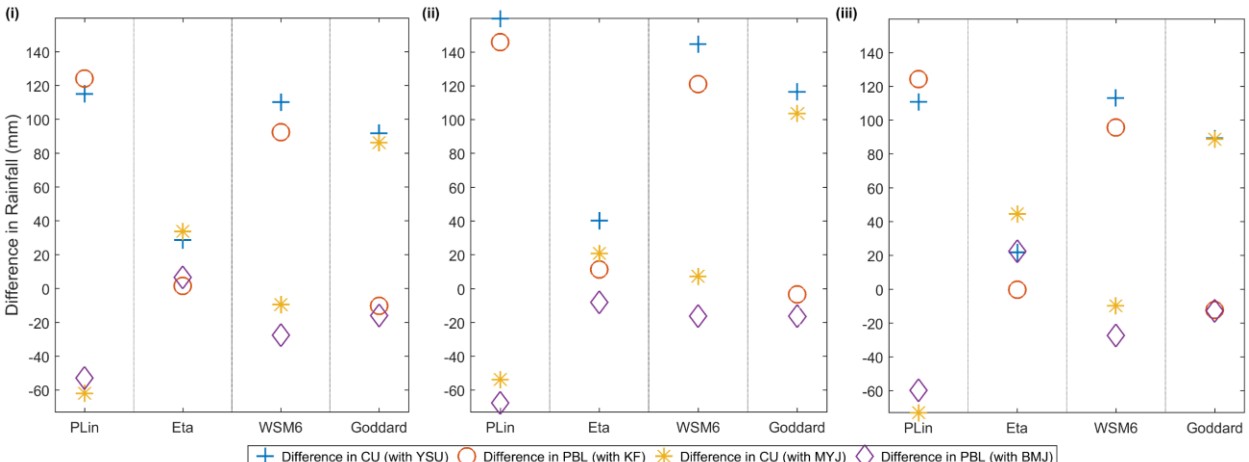

**Figure 10.** Difference in simulated rainfall due to PBL, CU and MP parametrization schemes corresponding to (i) Domain 1; (ii) Domain 2a; and (iii) Domain 2b over the UGB region.

It is noticed that, in general, the WRF configurations with KF convective scheme produce rainfall of higher magnitudes. This result is consistent with previously conducted studies (Gallus Jr, 1999; Fonseca et al., 2015; Pieri et al., 2015). For PLin MP, it is noted that considering YSU PBL along with KF CU scheme has a synergistic effect, leading to the maximum amount of rainfall over the region. This additive effect could be attributed to the YSU being a non-local scheme making it suitable for convective, unstable PBL conditions (Bright and Mullen, 2002). Upon changing the PBL scheme (from YSU to MYJ), and maintaining the convective scheme as KF, notable difference in the fields is simulated (as shown by red circles in Fig. 10). This difference obtained for changing the PBL (with PLin MP and KF CU), is found to be equivalent to the case when only the CU is changed (from KF to BMJ) under YSU PBL scheme (shown by blue plus sign in Fig. 10). Similarly, the difference in rainfall obtained for two cases –

changing the PBL (from YSU to MYJ) with PLin MP and BMJ CU; and changing the CU (from KF to BMJ) with PLin MP and MYJ PBL is also found to be approximately equal. This indicates that the average cumulative rainfall values obtained under two configurations – PLin MP, BMJ CU, and YSU PBL and PLin MP, KF CU, and MYJ PBL are almost equal. Further, BMJ CU, irrespective of the PBL scheme, results in less simulated rainfall across the region. For WSM6 MP, within YSU PBL scheme, changing the CU (from KF to BMJ) parametrization produces

significant variability (displayed by blue plus sign) in rainfall than changing the PBL scheme itself (from YSU to MYJ PBL with KF CU). However, with MYJ PBL, the effect of changing CU scheme is insignificant (yellow star in Fig. 10). Furthermore, with BMJ the difference in rainfall produced due to changing the PBL is minimal. With Eta and Goddard MP, changing the PBL (irrespective of the CU scheme) produces a negligible difference in the rainfall simulations, as represented by red circles and purple diamonds in Figure 10. However, changing the CU schemes

(irrespective of the PBL condition) is seen to have a significant influence on rainfall. It can be concluded from this section that the relationship between PBL and CU is interlinked, wherein YSU and MYJ PBLs complement (contradicts) the effect of KF (BMJ) and BMJ (KF) CUs respectively on the quantity of rainfall. Overall, the choice of CU appears to have a significant impact on the simulation of rainfall over the region. This conclusion is consistent with the earlier studies such as by Sikder and Hossain, (2016), where they found ISMR to be more

sensitive to CU than to MP.

**3.1.3 Impact of Land Surface Boundary Condition**

It is well established that the soil moisture plays a significant role in weather predictions (Chen and Brutsaert, 1995; Betts et al., 1996; Entekhabi et al., 1996.; Betts et al., 1997). Therefore, Slab and Noah LSMs, which differ significantly in two factors – (i) the soil depths along with the inclusion of land surface processes and (ii) the

temporal evolution of soil moisture, are selected in the present study to assess the impact of land surface conditions on simulation of heavy rainfall events. The Slab is a relatively simple LSM with 5 soil layers (at 1, 2, 4, 8 and 16 cm depths) and uses a thermal diffusion equation to compute surface fluxes based on a surface temperature and drag coefficient formulations. Noah LSM is modestly detailed (compared to Slab) LSM with 4 soil layers (at 10, 30, 60 and 100 cm depths) and explicit representation of land surface parameters, which includes the effect of soil moisture

changes, snow cover, evapotranspiration and hydrologic processes such as runoff and drainage (to subsurface layers). Further, in the Noah LSM soil moisture and temperature is prognostically computed for each of the 4 soil layers, whereas in the Slab LSM only soil temperature is prognostic and moisture is considered as a constant value based on the land-use. Slab lacks the feature of predicting the snow cover and does not capture the evaporation and runoff processes over the region.

To understand the influence of each LSM on the rainfall estimation, simulations using the Slab LSM are conducted for the 'best' and the 'worst' performing configurations from the Noah LSM case (configuration (p) and (b), respectively). Results comparing the two land model based runs are presented in Figure 11. The top panel in Figure 11 presents the scatter plot of the cumulative rainfall obtained with Noah and Slab LSM runs versus the observed cumulative rainfall. Bottom panel (Fig. 11) presents the mean difference in the simulated and the observed rainfall

values over the 4 days' period for 18 locations along with the spatially averaged values (last column in the bottom panel) within the region.

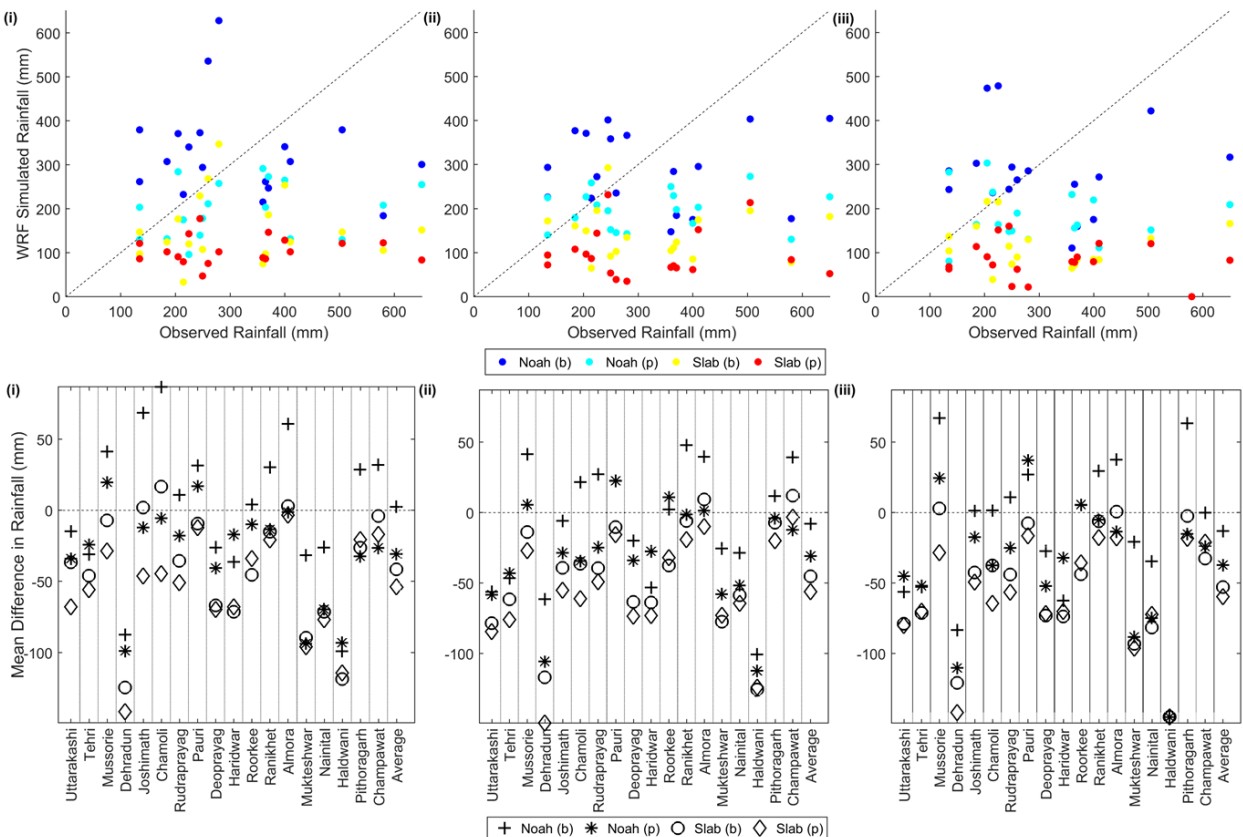

**Figure 11.** Scatter plot (top panel) and mean difference in rainfall (bottom panel) for the observed rainfall data and the WRF simulations (for (b) and (p) configurations) pertaining to Noah and Slab LSMs corresponding to (i) Domain 1; (ii) Domain 2a; 460  and (iii) Domain 2b.

The Slab LSM based run significantly underestimates the rainfall in comparison to the Noah LSM. For example, the locations which recorded rainfall greater than 400 mm have the Slab LSM based simulated values in the range of 100 – 150 mm. As stated earlier, although Noah LSM also underestimated the rainfall for such stations, the bias with the Noah LSM is significantly less than the Slab LSM (-26 % with the Noah LSM, in contrast to -64 % with the Slab 465  LSM for domain 2a and (p) configuration). Further, the mean difference in rainfall obtained with Slab LSM is found to be higher in comparison to the Noah LSM. This is essentially due to significant underestimation of rainfall during 16 and 17 June 2013 by the Slab LSM.

In a number of studies, the differences in the surface energy fluxes simulated by the choice of different LSMs i.e. Slab versus Noah has been discussed (see (Niyogi et al., 2016) for a review). The main reason being that the surface 470  processes affect the boundary layer feedbacks which in turn create zones of mesoscale convergence that can affect the location and intensity of convection. These convective systems then contribute to the simulated rainfall. The results obtained in this study emphasize this feature with differences in the rain amounts and locations in response to the change in LSM. The better performance of using the Noah model could be attributed to the temporal evolution of soil moisture fields. Analyzing the soil moisture in Slab and Noah model, the soil is noted to be relatively dry in 475  Slab (soil moisture less than 0.05 $m^3/m^3$) and the value is constant throughout the model run (since in the Slab model

there is no prognostic soil moisture term). In case of Noah, soil moisture varies in response to the rainfall and is found to vary between 0.25 m³/m³ to 0.45 m³/m³. Higher surface moisture conditions improve mass flux, convective updrafts and diabatic heating in the boundary layer that contributes to low level positive potential vorticity or convective potential which leads to enhanced rainfall potential (Osuri et al., 2017a). The importance of representing soil moisture variability over India for extreme weather conditions is also highlighted through this work.

### 3.2 Comparison between Rainfall from the WRF and the FNL dataset

Simulated rainfall from the WRF model runs is assessed with respect to the NCEP FNL reanalysis dataset. To achieve this, it is necessary to bring both the datasets to a common spatial resolution. Therefore, the WRF simulated rainfall is upscaled, through averaging of the grids, to match the resolution of NCEP FNL (1°×1°). For the analysis, simulations pertaining to the 'best' performing configuration (p), are only considered. Bias ($\beta$) in rainfall simulations from the two datasets corresponding to 18 rain gauge locations is obtained, results for which are presented in Figure 12.

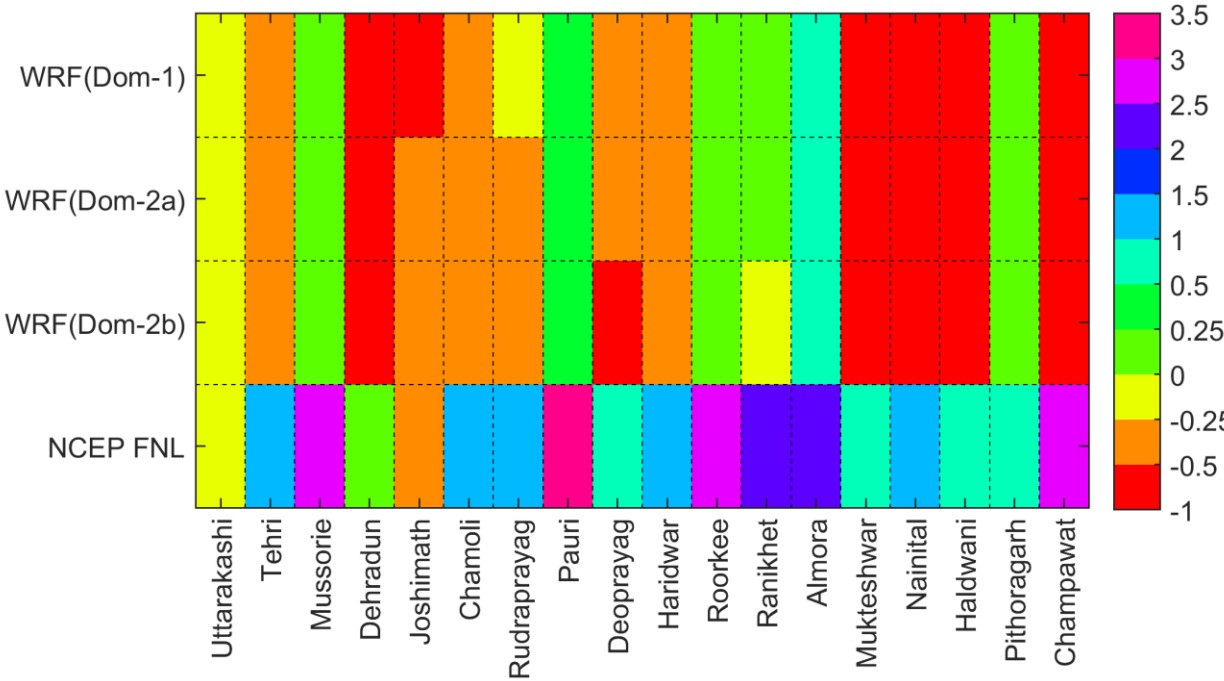

**Figure 12.** Bias ($\beta$) in rainfall simulations obtained from the NCEP FNL and WRF (upscaled to 1°×1°) data.

From Figure 12 it can be observed that the NCEP FNL data overestimates the rainfall for most of the locations. Upon dynamic downscaling of the FNL data through the WRF model, rainfall simulations improved over the UGB region. Locations such as Mussorie, Pauri, and Roorkee, which have shown $\beta$ between 2.5 to 3.5 in the FNL data, reduced to 0 to 0.25 in the WRF simulations. Uttarakashi and Pithoragarh locations having a small bias in the FNL data show similar small bias in the WRF simulations. Dehradun along with three stations from the south-eastern region, (Mukteshwar, Haldwani, and Nainital), which recorded heavy rainfall (Section 2.1), are observed to have a small bias in the FNL data, and the rainfall at this location is underestimated by the WRF model. Overall, rainfall simulations from the WRF model (for all the three domains) have less $\beta$ compared to the FNL data even after

upscaling to the resolution of 1°×1°. As expected, upon upscaling, the spatial variability between the domains is reduced due to averaging across several grid points.

From the above analysis, it is evident that the WRF model can simulate extreme precipitation better than the reanalysis data. This can be attributed to increase in spatial resolution, and better representation of surface and meteorological features, with respect to the lateral boundary conditions as suggested in some of the previous works such as by Argüeso et al., (2011); Mishra et al., (2014); Giorgi and Gutowski Jr, (2015); and Singh et al., (2017).

## 4. Summary and Conclusions

The main focus of this paper is to provide a general guideline for setting up the WRF model configuration to simulate heavy rainfall events. In this regard, sensitivity of the WRF model to predict heavy to extremely heavy rainfall events is examined through (a) quantitative verification of the rainfall simulated by the WRF model; (b) investigating sensitivity of the simulated rainfall to different parametrization schemes, downscaling ratios, and land surface models; (c) testing the selected scheme for other rainfall events; and (d) assessing the effect of local and
global factors by comparing the simulated rainfall with global reanalysis dataset.

For the analysis, an extremely heavy rainfall event, which occurred from 15 to 18 June 2013, over the Ganges basin, in the foothills of the Himalayas in the Uttarakhand State of northern India is considered. Most of the studies conducted earlier over this region (Medina et al., 2010; Kumar et al., 2012; Thayyen et al., 2013; Kumar et al., 2014; Chevuturi et al., 2015; Shekhar et al., 2015; Chevuturi and Dimri, 2016; Rajesh et al., 2016; Hazra et al., 2017) are
based on the general/default WRF configuration with WSM 6 microphysics, Kain-Fritsch cumulus parametrization scheme and planetary boundary layer of Yonsei University scheme. In this paper, ensemble experiments are conducted using the WRF model with different grid spacing, four microphysics schemes, two cumulus parametrization schemes, two planetary boundary layer schemes and two land surface model conditions. The rainfall simulations are evaluated against the observed rain gauge data and the TMPA precipitation data. The WRF
configuration with Goddard microphysics, Mellor–Yamada–Janjic planetary boundary layer condition and Betts–Miller–Janjic cumulus parameterization scheme is found to perform 'best' in simulating extremely heavy rain event of June 2013. The selected configuration is then verified by simulating several other heavy to extremely heavy rainfall events that occurred across different months in the monsoon season over the upstream region of the UGB. The results for the additional events indicate that the selected configuration (Goddard microphysics, Mellor–
Yamada–Janjic planetary boundary layer condition and Betts–Miller–Janjic cumulus parameterization scheme) is indeed the 'best' in simulating the spatial and temporal variability of the extremely heavy rainfall over the region. Therefore, through the exhaustive analysis conducted in this paper, the recommended WRF configuration for extreme rainfall simulations in the Himalayan region is Goddard microphysics, Mellor–Yamada–Janjic planetary boundary layer condition and Betts–Miller–Janjic cumulus parameterization scheme.
Although complex interactions are observed between different physics options, microphysics schemes are noticed to influence the spatial pattern of the rainfall, while the choice of cumulus scheme is found to modulate the magnitude of the simulated rainfall. Upon analyzing the impact of downscaling ratios on rainfall simulations, it is concluded that downscaling from global to regional scale with moderate downscaling ratio may give least model errors and

thus, be considered as suitable for reproducing the extreme rainfall event. In addition to this, the effect of land surface models (LSMs) on rainfall simulations is also assessed in this paper. The Slab LSM significantly underestimates the rainfall values, and incorporating Noah helped improve the performance. The underperformance of the Slab model is attributed to dry soil conditions in the region for this LSM.

In addition to the sensitivity experiments, the WRF simulated rainfall is also compared with the NCEP FNL reanalysis data. The NCEP FNL data is found to overestimate the rainfall whereas, the WRF simulated rainfall exhibited less bias. The comparison results indicated that care must be taken while employing global datasets for regional analysis. Through this, it can be established that the rainfall values obtained from the high-resolution mesoscale model can be effectively used in hydrologic models for realistic streamflow estimates.

The analyses presented in this paper are subjected to a few limitations: first, results are limited to the physics parametrization schemes considered in this paper, and may vary upon inclusion of other schemes; second, only two sets of downscaling ratios, i.e., 1:9 and 1:3 are tested in the current work. The sensitivity of simulations pertaining to other downscaling ratios should be tested in future; and third, only G2R and G2C sensitivity are assessed in this work.

**Acknowledgment**

The work is part of the study supported by the US National Science Foundation (CAREER AGS0847472) and the Government of India/MOES National Monsoon Mission (Grant no./Project no.: MM/SERP/CNRS/2013/INT-10/002) at Purdue University. The second author gratefully acknowledges the financial support of ESSO, Ministry of Earth Sciences, Government of India and Science and Engineering Research Board (SERB), Department of Science and Technology, Government of India.

**Appendices**

**Appendix A**

**Table A.1.** List of different WRF configuration

| WRF Configuration | Microphysics Scheme (MP) | Cumulus Scheme (CU) | Planetary Boundary Layer Scheme (PBL) |
|---|---|---|---|
| a | PLin | KF | YSU |
| b | Eta | KF | YSU |
| c | WSM6 | KF | YSU |
| d | Goddard | KF | YSU |
| e | PLin | BMJ | YSU |
| f | Eta | BMJ | YSU |
| g | WSM6 | BMJ | YSU |
| h | Goddard | BMJ | YSU |
| i | PLin | KF | MYJ |
| j | Eta | KF | MYJ |
| k | WSM6 | KF | MYJ |
| l | Goddard | KF | MYJ |
| m | PLin | BMJ | MYJ |
| n | Eta | BMJ | MYJ |
| o | WSM6 | BMJ | MYJ |
| p | Goddard | BMJ | MYJ |

## Appendix B

**Table B.1.** Mean Absolute Error (MAE) values corresponding to different WRF configuration for the three domains

| WRF Configuration | Domain 1 | | Domain 2a | | Domain 2b | |
|---|---|---|---|---|---|---|
| | Temporal | Spatial | Temporal | Spatial | Temporal | Spatial |
| a | 1028 | 228 | 943 | 210 | 1066 | 237 |
| b | 1227 | 273 | 1097 | 244 | 1249 | 278 |
| c | 960 | 213 | 1048 | 233 | 1077 | 239 |
| d | 825 | 183 | 852 | 189 | 908 | 202 |
| e | 877 | 195 | 885 | 197 | 995 | 221 |
| f | 1068 | 237 | 886 | 197 | 1055 | 234 |
| g | 887 | 197 | 950 | 211 | 979 | 218 |
| h | 885 | 197 | 976 | 217 | 1010 | 225 |
| i | 932 | 207 | 939 | 209 | 1024 | 228 |
| j | 1161 | 258 | 970 | 215 | 1131 | 251 |
| k | 894 | 199 | 888 | 197 | 987 | 219 |
| l | 883 | 196 | 855 | 190 | 918 | 204 |
| m | 890 | 198 | 913 | 203 | 974 | 217 |
| n | 1012 | 225 | 870 | 193 | 976 | 217 |
| o | 837 | 186 | 863 | 192 | 976 | 217 |
| p | 740 | 164 | 843 | 187 | 886 | 197 |

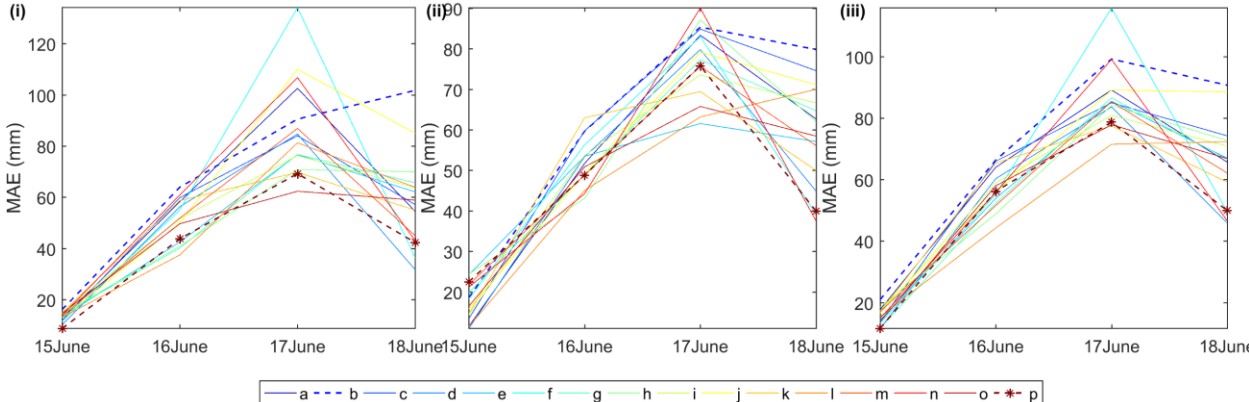

**Figure B.1.** Mean absolute errors in space corresponding to different WRF configurations for (i) Domain 1; (iii) Domain 2a; and (iii) Domain 2b. Blue dotted lines present the 'worst' performing configuration, i.e., configuration (b) and red dotted lines show the 'best' performing configuration, i.e., configuration (p).

**Appendix C**

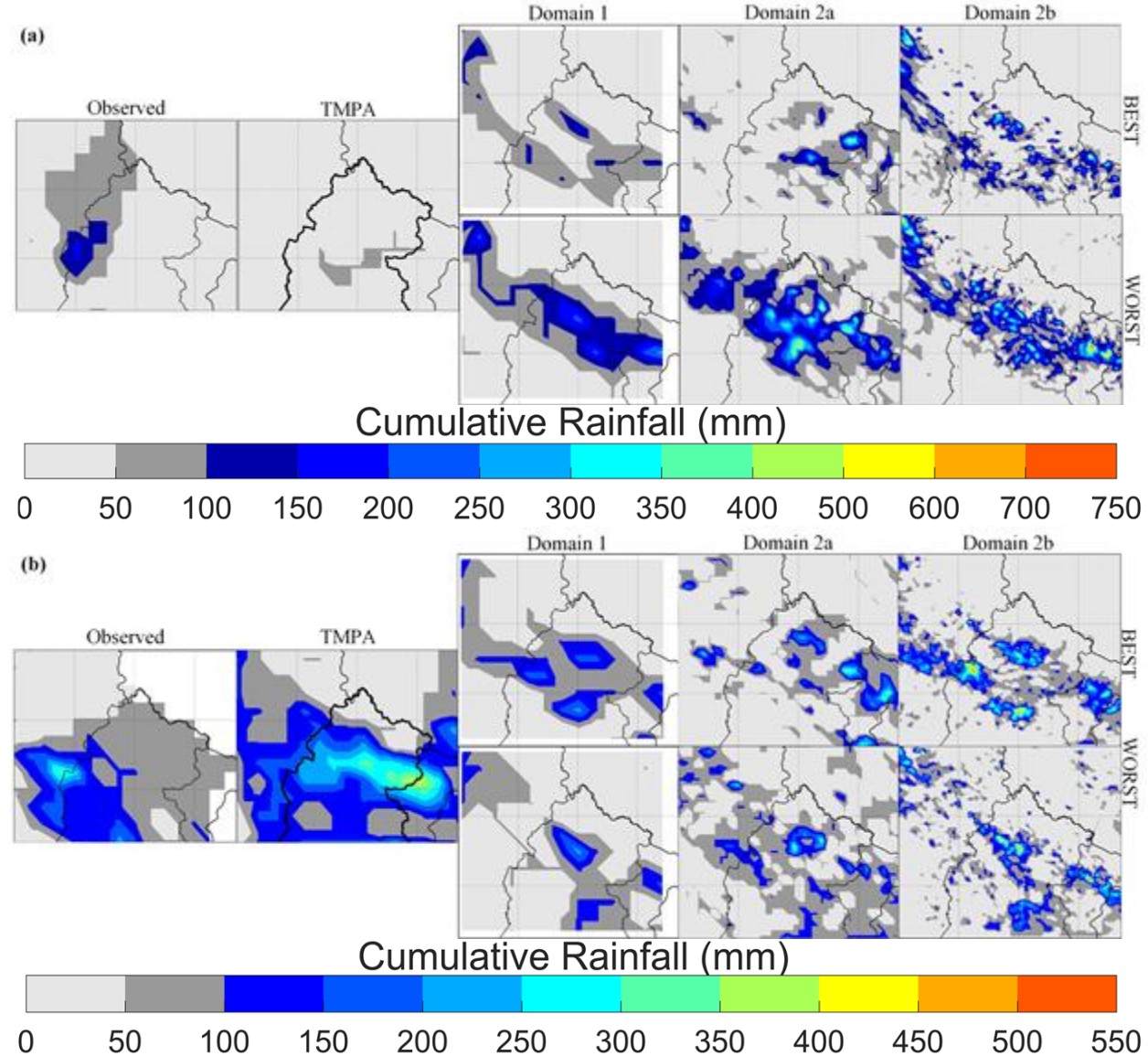


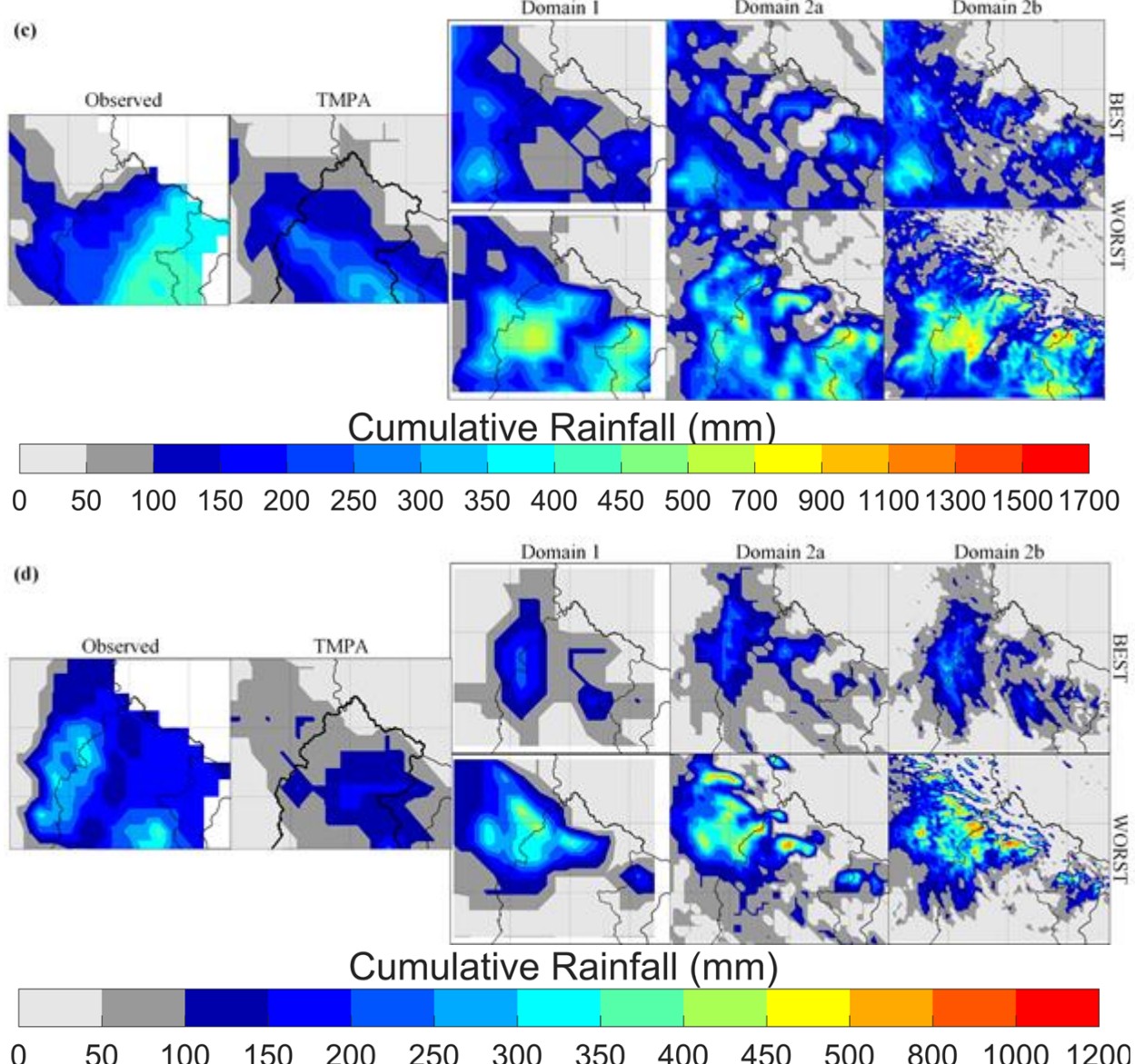

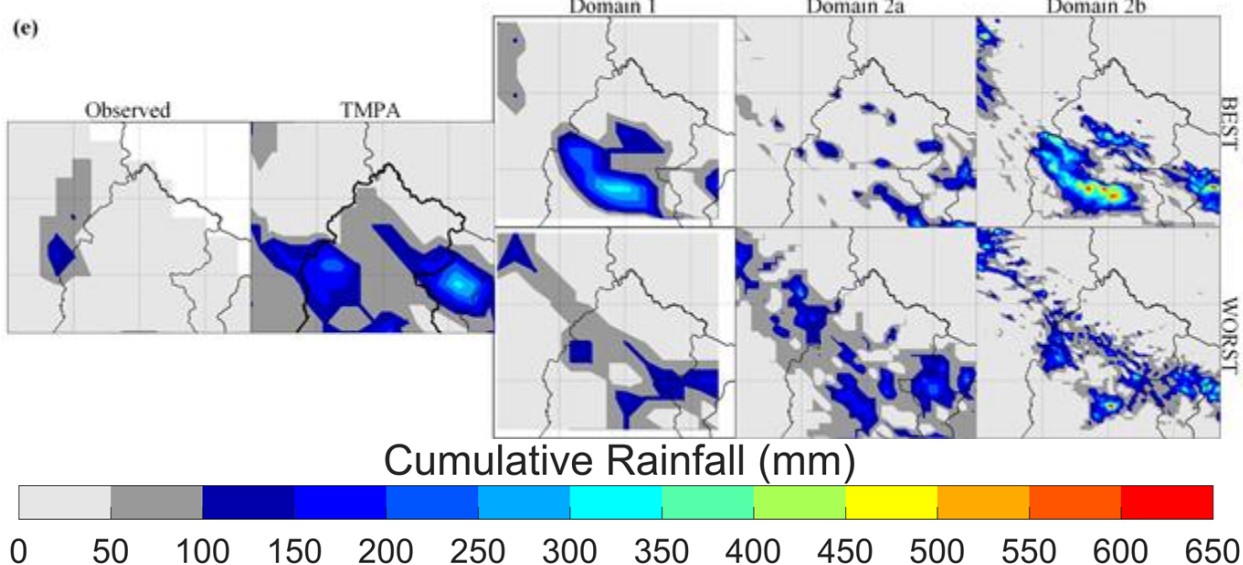

**Figure C.1.** Spatial plots presenting the rainfall simulations obtained across the three domains for the best and the worst configuration for heavy to extremely heavy rainfall events during (a) Event 1 (18 – 22 June 2008); (b) Event 2 (29 July – 2 August 2010); (c) Event 3 (15 – 19 August 2011); (d) Event 4 (17 – 21 September 2010); and (e) Event 5 (11 – 15 September 2012).


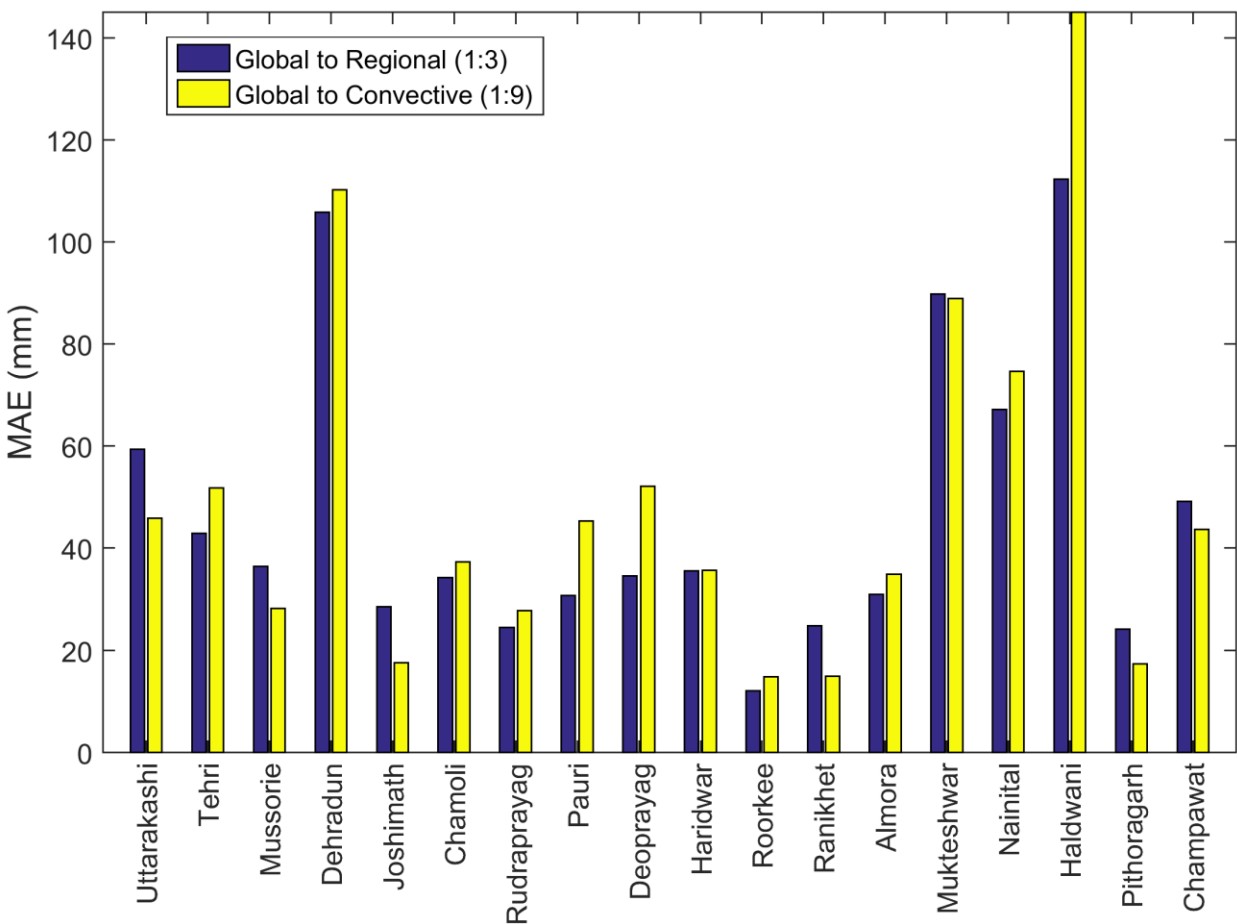

**Figure D.1.** Bar plot representing the mean absolute errors in simulating rainfall across the 18 rain gauge locations at Global to Regional (G2R) scale (1:3) and Global to Convection-permitting (G2C) scale (1:9).

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
