# Peer review of "Assessment of the Weather Research and Forecasting (WRF) Model for Simulation of Extreme Rainfall Events in the Upper Ganga Basin"

_Hydrology and Earth System Sciences, 2017_

## Referee Comment (RC1) · Anonymous Referee #1 · 25 Sep 2017

The authors have used Weather Research and Forecasting model for extreme events over upper Ganga basin and evaluated the simulations. The work is of importance; however, there are certain comments that need to be addressed.

1. I have reservation in stating " However, setting up the WRF model, that simulates extremely heavy rainfall over the ISMR region is still considered as a challenging task..". In my opinion setting up WRF is no longer a challenging task, given multiple works have been reported on the same. However, finding the best physics parameterization option or understanding of the combinations of good parameterization options for different purposes is still an area of research and that needs to come out through the first

paragraph of introduction.

2. I also have reservation in selecting an extreme event without understanding how does the regional model work for seasonal monsoon rainfall over the region. Do they add value to the simulations by global models? What about the existing literature on evaluation of CORDEX in adding values? Which one is more sensitive, microphysics parameterization or cumulative parametrization. How does WRF perform in different years, dry, wet or normal years? There are multiple works that have been published recently. The authors need to perform a good review of recent literature, identify the gap and define the problem. This is missing in the present version of the manuscript.

3. The authors need to present the evaluation of the regional model at least for one season of monsoon (for all 122 days). We have to make sure that the selected parameterization does not overestimate for all the days and hence performing well for the extreme days. This simulation needs to be performed.

4. Figure 4 is wrongly interpreted. The CORDEX models have the boundary conditions from CMIP5 models that do not have any observed initial condition. Hence, it is not correct to pick up specific dates from the simulations and compare. I think it is better to delete this figure.

5. Similarly Figure 12 also has the same problem if the bias is for those specific days.

6. I would specifically suggest to delete the CORDEX part, as it may not be directly related to the work (if authors want they may pick up the evaluation runs that are forced with reanalysis data, but such simulations may not be available for 2013). They should focus on identifying the added value by regional model in comparison with the reanalysis data that is being used as boundary condition.

7. I am not very sure, if the use of single extreme is sufficient for any conclusion.

8. I also would like to know the role of land surface processes in this extreme event. Some details on the land surface module that has been coupled to WRF, may also be

useful.

---

## Referee Comment (RC2) · Anonymous Referee #2 · 9 Oct 2017

The paper provides results of a case study for an extreme event in northern India using WRF with multiple physics combinations. It evaluates rainfall primarily against observational data at stations and derived from TRMM. This focuses on the heavy rain period of 15-18 June 2013, and therefore conclusions about performance of the different simulations cannot be taken to be very robust in their usefulness for other cases. This limits the usefulness of this study. While it is interesting that certain physics combinations performed well and some parameterizations did well in several different combinations, it was not clearly presented.

Some physics schemese showed greater sensitivity to other schemes combined with

them, as seen in Fig. 10 for example. But showing the absolute difference hides some information that could have been seen from the difference itself that may have negative values.

It is also hard to determine signal from noise in Figures 7,8, and 11 when a station mean might have been of value in showing overall trends.

A major problem I have is with the use of CORDEX data. CORDEX is downscaled from climate model simulations and would therefore not be expected to bear any resemblance to real weather on any specific date. These are not comparable with weather models driven by real anaysis boundary conditions. Sampling CORDEX on a particular date of a heavy rainfall event therefore will not be a fair comparison because it is more likely to miss such events entirely while it may have them on other days depending on which global data is used. It is no surprise that CORDEX runs underestimate heavy events on a particular date, which does not mean they underestimate them in general. Such data can only be used qualititively to see if they can capture heavy events over many years that they are run using frequency analyses. I therefore suggest that the CORDEX part serves no value for this case study, and if the authors are using CORDEX it can only be in the context of the climatology of heavy events and whether the observed peak can be captured at other times of those runs. Maybe these runs never have such events, which may be useful to know, or maybe they have them too frequently, also useful.

Specific Points

1. line 222. It said KF is shallow convection when it has both deep and shallow convection.

2. line 231. Both PLin and WSM6 are 6-class if vapor is included as a class.

3. Figure 5. It is noted that domain 2b has no cumulus scheme within the domain, yet shows sensitivity to cumulus schemes, presumably through its boundaries and parent

domain.

4. Figure 6. Over what period is this rainfall summed? Is it interpolated to the station point?

5. line 288. Presumably the complex terrain is also a factor in the bias at stations. Even at 3 km, there may be flow and rainfall differences because the model does not fully resolve all the terrain details.

6. Figure 7. Is this the MAE for the total 4-day precipitation at each station?

7. Care should be taken when suggesting Goddard is best especially as it has less overall precipitation. Lower precipitation itself may lead to lower absolute errors than schemes that more correct total amounts in the wrong places. Smoother precipitation fields may always score favorably in MAE. Total precip is an important factor to evaluate.

8. line 316. BMJ even does better in 2b where it is not used and only would contribute through the boundaries, and this is surprising.

9. line 354. SE has not been defined. It looks like a ratio of model to observed variance.

10. Figure 9. It is confusing that colors are used both for rainfall and CV. Perhaps rainfall can be contoured.

11. line 416-424. SLAB underestimates rainfall. This raises the issue of its moisture availability value in this region. How high is it? Can a higher value give a better rainfall?

12. Major issue with using CORDEX as it is. See above comments.

13. Major issue with conclusions being drwan from one case. See above.

---

## Author Comment (AC1) · 25 Nov 2017

**Authors' Response to the Reviewer Comments**

**Manuscript Ref.: hess-2017-533**

**Title: Assessment of the Weather Research and Forecasting (WRF) Model for Extreme Rainfall Event Simulations in the Upper Ganga Basin**

**Authors:** Ila Chawla, Krishna K. Osuri, Pradeep P. Mujumdar, and Dev Niyogi

We sincerely thank the reviewers for their comments on the manuscript and offering their suggestions and critical input that has helped improve the manuscript. We provide here our replies to the reviewers' comments and highlight the changes made in the revised manuscript based on the comments. Sections which are modified in the revised manuscript are mentioned in this document.

**Responses to the comments of Referee #1:**

**General Comments:**

The authors have used Weather Research and Forecasting model for extreme events over upper Ganga basin and evaluated the simulations. The work is of importance; however, there are certain comments that need to be addressed.

**Comment 1:** I have reservation in stating " However, setting up the WRF model, that simulates extremely heavy rainfall over the ISMR region is still considered as a challenging task..". In my opinion setting up WRF is no longer a challenging task, given multiple works have been reported on the same. However, finding the best physics parameterization option or understanding of the combinations of good parameterization options for different purposes is still an area of research and that needs to come out through the first paragraph of introduction.

**Response:** We agree and have modified the statement. The statement was meant to highlight that it is a challenging task since it involves consideration of several aspects such as forcing data, model grid spacing/resolution, land surface parameterization and choice of an appropriate physics scheme.

**Action:** Lines "However, setting up the WRF model, that simulates extremely heavy rainfall over the ISMR region is still considered as a challenging task, which involves consideration of several aspects such as forcing data, model grid spacing/resolution, land surface parameterization and

choice of an appropriate physics scheme." are modified in the revised manuscript and now read as:

"*However, finding the optimal set of physics parameterization schemes (along with the selection of an appropriate model grid spacing/resolution) to simulate extreme/heavy rainfall events, and understanding the effect of the combination of different parametrization schemes on rainfall estimates over the Indian monsoon region are still an active area of research.*"

**Comment 2:** I also have reservation in selecting an extreme event without understanding how does the regional model work for seasonal monsoon rainfall over the region. Do they add value to the simulations by global models? What about the existing literature on evaluation of CORDEX in adding values? Which one is more sensitive, microphysics parameterization or cumulative parametrization. How does WRF perform in different years, dry, wet or normal years? There are multiple works that have been published recently. The authors need to perform a good review of recent literature, identify the gap and define the problem. This is missing in the present version of the manuscript.

**Response:**

*'How does the regional model work for seasonal monsoon rainfall over the region? Do they add value to the simulations by global models?'*

Global models have been employed in several studies to understand the large-scale circulation pattern and for quantitative analysis of the monsoon rainfall, but due to their coarse resolution, they are unable to represent the local to regional characteristics of monsoon rainfall. Regional models, on the other hand, can explicitly simulate the interactions between the large-scale weather phenomenon and regional topography, making the climate simulations reliable (Ratna et al., 2011; Wang et al., 2005; Kang et al., 2002; Gadgil et al., 2005; Srinivas et al., 2013). Furthermore, regional models have a better representation of convection thus offsetting one of the major sources of errors and uncertainties in the global models. Therefore, mesoscale models, such as the Weather Research and Forecasting (WRF) model, becomes a preferred choice to study seasonal monsoon rainfall.

The WRF model has been used as a diagnostic tool to understand the Indian Summer Monsoon Rainfall (ISMR) over the Himalayan region. For example, Kumar et al., (2012) used the WRF model to simulate the cloudburst event of 2010 in the Leh area over the north-western Himalayan

belt. While, Kumar et al., (2014) and Thayyen et al., (2013) used the WRF model to gain insight into the atmospheric processes and the mesoscale convective system (MCSs) that led to the 2010 Leh event. Similarly, Chevuturi et al., (2015) simulated the heavy precipitation event of September 2012 in the central Himalayas using the WRF model. Medina et al., (2010) used the WRF model to understand how topography and land surface conditions affect the extreme convection in western and eastern Himalayas. Particularly for the 2013 heavy rainfall episode in the Uttarakhand region, the WRF model is used in several studies, including those by Kotal et al., (2014); Vellore et al., (2016); and Hazra et al., (2017) to understand the physical processes leading to the event. Shekhar et al., (2015); Dimri et al., (2016); and Chevuturi and Dimri, (2016) performed in-depth synoptic and mesoscale analysis of the June 2013 heavy rainfall event using the WRF model. Rajesh et al., (2016) presented the role of land surface conditions in simulating the heavy rainfall event. Therefore, from the existing literature, it can be established that the regional model performs considerably well over the region.

Although model analysis of the June heavy rainfall event in the Uttarakhand state has been studied, ensemble analysis emphasizing the impact of the interaction between different model configurations in simulating the heavy rainfall event, and the associated variability (uncertainty) is still lacking. With this perspective, this paper seeks to assess the sensitivity of the WRF model to predict extremely heavy rainfall events.

*'What about the existing literature on evaluation of CORDEX in adding values?'*

In the earlier submitted version of the manuscript, we attempted to analyze the extreme rainfall event from the CORDEX data and observed that the rainfall is significantly underestimated by the CORDEX products (Section 2.2, old manuscript). Based on the comments received from both the reviewers and the focus we wish to keep in this study – which is the ability and the sensitivity/variability within WRF runs for simulating the heavy rain event(s) – we have now eliminated the analysis related to the CORDEX data in the revised manuscript. Nonetheless, with regards to the existing literature on this topic, Ali et al., (2014) studied the extreme rainfall projected by the CORDEX RCMs over the urban areas in India. They observed CORDEX-RCMs have a significant bias in the monsoon maximum rainfall, which could be attributed to model parametrization (Gutowski Jr et al., 2010) and model resolution (Wehner et al., 2010; Tripathi and Dominguez, 2013).

*'Which one is more sensitive, microphysics parameterization or cumulative parametrization?'*

The experiment results indicate that the microphysics parameterization and cumulative parametrization work in tandem in simulating the rainfall. The former appears to influence the spatial pattern of the rainfall better, while the convective parameterization influences the quantity of rainfall (Section 3.1.2). However, the results from this study alone, and the interdependency of the two aspects limit the ability to ascertain whether the simulated rainfall is 'more' sensitive to which of the two parameterization processes.

*'How does WRF perform in different years, dry, wet or normal years?'*

Since the main aim of the paper is to assess the sensitivity of the WRF model to simulate heavy rainfall events and understand the effect of the combination of different parametrization schemes, the performance of the WRF model for dry, wet or normal years deemed not to be a major concern. **Action:** The introduction section is significantly modified and additional literature is now added in the revised manuscript.

**Comment 3:** The authors need to present the evaluation of the regional model at least for one season of monsoon (for all 122 days). We have to make sure that the selected parameterization does not overestimate for all the days and hence performing well for the extreme days. This simulation needs to be performed.

**Response:** This comment was again more aligned with the CORDEX part of the study – which has been taken off from the revision. In the study, we are using a mesoscale model configuration to obtain the reliable forecasts for which typically 3- 5 days period is considered. Therefore, to run the model for entire monsoon season (i.e. 122 days), the model needs to be reinitialized after every few days, wherein boundary conditions are obtained from NCEP FNL reanalysis dataset while the initial conditions are obtained from the forecasted data of the previous cycle. This process is computationally expensive, and not central to the study goal (as revised now), nor is it feasible under the current model set-up. Nonetheless, we appreciate the point the reviewer is making and as an alternative approach to test the performance for more than one setting, we have identified additional cases of heavy to extremely heavy rainfall events over different months of the monsoon season and conducted model experiments for these additional events to access the model performance. The configuration with MYJ PBL, BMJ CU, and Goddard MP is again found to perform 'best' in simulating the spatial and temporal variability of the extremely heavy rainfall over the upstream region of the UGB – adding more credence and generality to the study findings.

**Action:** Results pertaining to the additional heavy to extremely heavy rainfall events are added to the revised manuscript (Section 3.1.1). Please refer to the response to comment 7 (Referee #1) for further details.

**Comment 4:** Figure 4 is wrongly interpreted. The CORDEX models have the boundary conditions from CMIP5 models that do not have any observed initial condition. Hence, it is not correct to pick up specific dates from the simulations and compare. I think it is better to delete this figure.

**Response:** The analysis related to the CORDEX data is deleted from the revised manuscript.

**Action:** Figure 4 (old manuscript) presenting the variability in daily and cumulative rainfall obtained from the CORDEX downscaled data is removed from the revised manuscript.

**Comment 5:** Similarly Figure 12 also has the same problem if the bias is for those specific days.

**Action:** Figure 12 (b) (in the old manuscript) presenting the comparison of CORDEX data with the WRF simulations is removed in the revised manuscript.

**Comment 6:** I would specifically suggest to delete the CORDEX part, as it may not be directly related to the work (if authors want they may pick up the evaluation runs that are forced with reanalysis data, but such simulations may not be available for 2013). They should focus on identifying the added value by regional model in comparison with the reanalysis data that is being used as boundary condition.

**Response:** We agree.

**Action:** CORDEX section (Section 2.2 (old manuscript) and part of Section 3.2) is removed from the revised manuscript.

**Comment 7:** I am not very sure, if the use of single extreme is sufficient for any conclusion.

**Response:** We agree, and simulations for five different (additional) heavy to extremely heavy rainfall events, each corresponding to the individual month of the monsoon season (June to September), that occurred in the upstream region of the UGB are now included in the revised manuscript.

**Action:** Following details are added in the revised manuscript:

Section 2.1:

*"In addition to June 2013 case, five additional heavy to extremely heavy rainfall events are also considered in the present study for the analysis, details of which are presented in Table R.1. Rainfall from the IMD gridded data at 0.25° resolution (Pai et al., 2014) is considered as the observed data for these events.*

**Table R.1.** *Heavy to extremely heavy rainfall events recorded in the UGB region*

| Event No. | Time Period | Maximum Rainfall Day | Maximum Rainfall Amount (mm) |
|-----------|-------------|----------------------|------------------------------|
| 1 | 18 – 22 June 2008 | 20 June | 126 |
| 2 | 29 July – 2 August 2010 | 31 July | 271 |
| 3 | 15 – 19 August 2011 | 16 August | 234 |
| 4 | 17 – 21 September 2010 | 19 September | 218 |
| 5 | 11 – 15 September 2012 | 14 September | 38 |

*It is to be noted that on 13 – 14 September 2012, cloudburst event was reported in the region and the total amount of rainfall on 14 September was recorded approximately to be 210 mm (Chevuturi et al., 2015). This event is significantly underestimated in the IMD gridded data, indicating that caution must be exercised while using the data for applications involving heavy rainfall events, such as flood modeling and validating the rainfall simulations from the mesoscale models. Figure R.1 presents the spatially averaged daily and cumulative rainfall received during different events (as specified in Table R.1).*

[Figure]

**Figure R.1.** *Spatially averaged daily and cumulative rainfall for Event 1 (18 – 22 June 2008); Event 2 (29 July – 2 August 2010); Event 3 (15 – 19 August 2011); Event 4 (17 – 21 September 2010); and Event 5 (11 – 15 September 2012) in the upstream region of the UGB.*

Section 3.1.1:

*To further assess the sensitivity of configuration (p) and configuration (b) in capturing the extreme rainfall events in the region, additional simulations pertaining to other heavy to extremely heavy rainfall events (as mentioned in Table R.1) are conducted. Spatial plots showing the cumulative rainfall estimates obtained for the three domains in comparison to the observed IMD gridded data and the TMPA data are presented in Appendix C. To summarize the performance of configuration (p) and configuration (b) against the observations (IMD gridded data), spatio-temporal MAE values are computed, which are presented in Table R.2.*

**Table R.2.** *Mean Absolute Error (MAE) values (in mm) corresponding to WRF configuration (p) and configuration (b) for the three domains*

| Event No. | Domain 1 | | Domain 2a | | Domain 2b | |
|---|---|---|---|---|---|---|
| | *(p)* | *(b)* | *(p)* | *(b)* | *(p)* | *(b)* |
| 1 | 10 | 13 | 10 | 14 | 11 | 14 |
| 2 | 18 | 23 | 18 | 23 | 21 | 22 |
| 3 | 39 | 45 | 38 | 44 | 40 | 46 |
| 4 | 23 | 28 | 23 | 28 | 24 | 29 |
| 5 | 12 | 12 | 9 | 13 | 12 | 11 |

*From the analysis conducted over the additional rainfall events, it is noted that configuration (p) gives less error in comparison to the configuration (b) for all the rainfall events. This makes configuration (p) with MYJ PBL, BMJ CU, and Goddard MP the 'best' in simulating the spatial and temporal variability of the extremely heavy rainfall over the upstream region of the UGB."*

Appendix C:

[Figure]

[Figure]

[Figure]

Cumulative Rainfall (mm)

[Figure]

***Figure R.2.*** *Spatial plots presenting the rainfall simulations obtained across the three domains for the best and the worst configuration for heavy to extremely heavy rainfall events during (a) Event 1 (18 – 22 June 2008); (b) Event 2 (29 July – 2 August 2010); (c) Event 3 (15 – 19 August 2011); (d) Event 4 (17 – 21 September 2010); and (e) Event 5 (11 – 15 September 2012).*

**Comment 8:** I also would like to know the role of land surface processes in this extreme event. Some details on the land surface module that has been coupled to WRF, may also be useful.

**Response:** Previous studies such as by Chang et al., (2009); Rajesh et al., (2016); Kishtawal et al., (2010); Lei et al., (2008); and Osuri et al., (2017) have focused exclusively on the impact of land surface characteristics in influencing the rainfall events. In particular, Rajesh et al., (2016) have discussed the role of land surface conditions on this particular event of heavy rainfall (June 2013 in Uttarakhand). They conducted two sets of experiment – one, without land data assimilation (referred as control experiment) and the other, with land data assimilation through utilization of high-resolution soil moisture and soil temperature in the WRF model (LDAS experiment). Their results indicate that the model accurately simulated the heavy rainfall in the LDAS case due to better representation of lower boundary conditions. The land model being used in the WRF configuration is the community Noah model and it has been cited.

Further, we have undertaken experiments to assess the effect of the land surface schemes on the rainfall simulations by considering two Land Surface Models (LSMs) – Noah LSM and the simple five-layer soil model (Slab). The Slab LSM based run resulted in a significant underestimation of the rainfall simulations in comparison to the Noah LSM. The better performance of using the Noah

model could be attributed to the temporal evolution of soil moisture fields. Additional details related to the land surface module used in the current study are added in the revised manuscript.

**Action:** Following paragraph is added in the revised manuscript (Section 2.2):

*"The sensitivity of various WRF configurations to simulate heavy rainfall events is assessed using the Noah LSM (Chen and Dudhia, 2001; Tewari et al., 2004; Ek et al., 2003). The Noah LSM is a community model that is included in the WRF suite with the prime aim of providing reliable boundary conditions to the atmospheric model. As a result, Noah LSM is moderately detailed model, which includes single canopy layer with canopy resistance scheme of Noilhan and Planton, (1989) and four soil layers (at 0.1, 0.3, 0.6 and 1.0 m) with a total soil depth of 2 m. The last soil layer of 1 m acts as a reservoir for drainage of water under gravity and the above three layers serve as root zone depths. There is a provision in the model to change default root zone depths with the actual values from the field, subjected to data availability. In the Noah LSM, surface (skin) temperature is obtained using a single linearized surface energy balance equation, which effectively considers the ground and vegetation surface. Frozen soil parametrization based on Koren et al., (1999) and surface runoff scheme of Schaake et al., (1996) are also included in this model. Soil moisture, soil temperature, water intercepted by the canopy and snow stored on the ground are also included as the prognostic variables in the model. More detailed information on the Noah LSM can be obtained from Ek et al., (2003).*

*To assess the effect of the land surface scheme on simulations, the Noah LSM is replaced with the simple five-layer Soil Model (Slab; (Dudhia, 1996)). In contrast to the relatively sophisticated Noah LSM, Slab is based on simple thermal diffusion in the soil layers that has constant soil moisture availability but a prognostic soil temperature term (Deardorff, 1978). Further differences between the two LSMs are presented in Section 3.1.3."*

**References**

Ali, H., Mishra, V., and Pai, D.: Observed and projected urban extreme rainfall events in India, Journal of Geophysical Research: Atmospheres, 119, 2014.

Chang, H.-I., Kumar, A., Niyogi, D., Mohanty, U., Chen, F., and Dudhia, J.: The role of land surface processes on the mesoscale simulation of the July 26, 2005 heavy rain event over Mumbai, India, Global Planet. Change, 67, 87-103, 2009.

Chen, F., and Dudhia, J.: Coupling an advanced land surface–hydrology model with the Penn State–NCAR MM5 modeling system. Part I: Model implementation and sensitivity, Mon. Weather Rev., 129, 569-585, 2001.

Chevuturi, A., Dimri, A., Das, S., Kumar, A., and Niyogi, D.: Numerical simulation of an intense precipitation event over Rudraprayag in the central Himalayas during 13–14 September 2012, J. Earth Syst. Sci., 124, 1545-1561, 2015.

Chevuturi, A., and Dimri, A.: Investigation of Uttarakhand (India) disaster-2013 using weather research and forecasting model, Nat. Hazards, 82, 1703-1726, 2016.

Deardorff, J.: Efficient prediction of ground surface temperature and moisture, with inclusion of a layer of vegetation, Journal of Geophysical Research: Oceans, 83, 1889-1903, 1978.

Dimri, A., Thayyen, R., Kibler, K., Stanton, A., Jain, S., Tullos, D., and Singh, V.: A review of atmospheric and land surface processes with emphasis on flood generation in the Southern Himalayan rivers, Sci. Tot. Environ., 556, 98-115, 2016.

Dudhia, J.: A multi-layer soil temperature model for MM5, Preprints, The Sixth PSU/NCAR mesoscale model users' workshop, 1996, 22-24,

Ek, M., Mitchell, K., Lin, Y., Rogers, E., Grunmann, P., Koren, V., Gayno, G., and Tarpley, J.: Implementation of Noah land surface model advances in the National Centers for Environmental Prediction operational mesoscale Eta model, Journal of Geophysical Research: Atmospheres, 108, 2003.

Gadgil, S., Rajeevan, M., and Nanjundiah, R.: Monsoon prediction—why yet another failure, Curr. Sci, 88, 1389-1400, 2005.

Gutowski Jr, W. J., Arritt, R. W., Kawazoe, S., Flory, D. M., Takle, E. S., Biner, S., Caya, D., Jones, R. G., Laprise, R., and Leung, L. R.: Regional extreme monthly precipitation simulated by NARCCAP RCMs, J. Hydrometeorol., 11, 1373-1379, 2010.

Hazra, A., Chaudhari, H. S., Ranalkar, M., and Chen, J. P.: Role of interactions between cloud microphysics, dynamics and aerosol in the heavy rainfall event of June 2013 over Uttarakhand, India, Q. J. Roy. Meteorol. Soc., 143, 986-998, 2017.

Kang, I.-S., Jin, K., Wang, B., Lau, K.-M., Shukla, J., Krishnamurthy, V., Schubert, S., Wailser, D., Stern, W., and Kitoh, A.: Intercomparison of the climatological variations of Asian summer monsoon precipitation simulated by 10 GCMs, Clim. Dyn., 19, 383-395, 2002.

Kishtawal, C. M., Niyogi, D., Tewari, M., Pielke, R. A., and Shepherd, J. M.: Urbanization signature in the observed heavy rainfall climatology over India, Int. J. Climatol., 30, 1908-1916, 2010.

Koren, V., Schaake, J., Mitchell, K., Duan, Q. Y., Chen, F., and Baker, J.: A parameterization of snowpack and frozen ground intended for NCEP weather and climate models, Journal of Geophysical Research: Atmospheres, 104, 19569-19585, 1999.

Kotal, S., Roy, S. S., and Roy Bhowmik, S.: Catastrophic heavy rainfall episode over Uttarakhand during 16–18 June 2013—observational aspects, Curr. Sci, 107, 234-245, 2014.

Kumar, A., Houze Jr, R. A., Rasmussen, K. L., and Peters-Lidard, C.: Simulation of a flash flooding storm at the steep edge of the Himalayas, J. Hydrometeorol., 15, 212-228, 2014.

Kumar, M. S., Shekhar, M., Krishna, S. R., Bhutiyani, M., and Ganju, A.: Numerical simulation of cloud burst event on August 05, 2010, over Leh using WRF mesoscale model, Nat. Hazards, 62, 1261-1271, 2012.

Lei, M., Niyogi, D., Kishtawal, C., Pielke Sr, R., BeltrĂĄn-Przekurat, A., Nobis, T., and Vaidya, S.: Effect of explicit urban land surface representation on the simulation of the 26 July 2005 heavy rain event over Mumbai, India, Atmos. Chem. Phys., 8, 5975-5995, 2008.

Medina, S., Houze, R. A., Kumar, A., and Niyogi, D.: Summer monsoon convection in the Himalayan region: Terrain and land cover effects, Q. J. Roy. Meteorol. Soc., 136, 593-616, 2010.

Noilhan, J., and Planton, S.: A simple parameterization of land surface processes for meteorological models, Mon. Weather Rev., 117, 536-549, 1989.

Osuri, K., Nadimpalli, R., Mohanty, U., Chen, F., Rajeevan, M., and Niyogi, D.: Improved prediction of severe thunderstorms over the Indian Monsoon region using high-resolution soil moisture and temperature initialization, Scientific Reports, 7, 2017.

Pai, D., Sridhar, L., Rajeevan, M., Sreejith, O., Satbhai, N., and Mukhopadhyay, B.: Development of a new high spatial resolution (0.25× 0.25) long period (1901–2010) daily gridded rainfall data set over India and its comparison with existing data sets over the region, Mausam, 65, 1-18, 2014.

Rajesh, P., Pattnaik, S., Rai, D., Osuri, K., Mohanty, U., and Tripathy, S.: Role of land state in a high resolution mesoscale model for simulating the Uttarakhand heavy rainfall event over India, J. Earth Syst. Sci., 125, 475-498, 2016.

Ratna, S. B., Sikka, D., Dalvi, M., and Venkata Ratnam, J.: Dynamical simulation of Indian summer monsoon circulation, rainfall and its interannual variability using a high resolution atmospheric general circulation model, Int. J. Climatol., 31, 1927-1942, 2011.

Schaake, J. C., Koren, V. I., Duan, Q. Y., Mitchell, K., and Chen, F.: Simple water balance model for estimating runoff at different spatial and temporal scales, Journal of Geophysical Research: Atmospheres, 101, 7461-7475, 1996.

Shekhar, M., Pattanayak, S., Mohanty, U., Paul, S., and Kumar, M. S.: A study on the heavy rainfall event around Kedarnath area (Uttarakhand) on 16 June 2013, J. Earth Syst. Sci., 124, 1531-1544, 2015.

Srinivas, C., Hariprasad, D., Bhaskar Rao, D., Anjaneyulu, Y., Baskaran, R., and Venkatraman, B.: Simulation of the Indian summer monsoon regional climate using advanced research WRF model, Int. J. Climatol., 33, 1195-1210, 2013.

Tewari, M., Chen, F., Wang, W., Dudhia, J., LeMone, M., Mitchell, K., Ek, M., Gayno, G., Wegiel, J., and Cuenca, R.: Implementation and verification of the unified NOAH land surface model in

the WRF model, 20th conference on weather analysis and forecasting/16th conference on numerical weather prediction, 2004,

Thayyen, R. J., Dimri, A., Kumar, P., and Agnihotri, G.: Study of cloudburst and flash floods around Leh, India, during August 4–6, 2010, Nat. Hazards, 65, 2175-2204, 2013.

Tripathi, O. P., and Dominguez, F.: Effects of spatial resolution in the simulation of daily and subdaily precipitation in the southwestern US, Journal of Geophysical Research: Atmospheres, 118, 7591-7605, 2013.

Vellore, R. K., Kaplan, M. L., Krishnan, R., Lewis, J. M., Sabade, S., Deshpande, N., Singh, B. B., Madhura, R., and Rao, M. R.: Monsoon-extratropical circulation interactions in Himalayan extreme rainfall, Clim. Dyn., 46, 3517-3546, 2016.

Wang, B., Ding, Q., Fu, X., Kang, I. S., Jin, K., Shukla, J., and Doblas-Reyes, F.: Fundamental challenge in simulation and prediction of summer monsoon rainfall, Geophys. Res. Lett., 32, 2005.

Wehner, M. F., Smith, R. L., Bala, G., and Duffy, P.: The effect of horizontal resolution on simulation of very extreme US precipitation events in a global atmosphere model, Clim. Dyn., 34, 241-247, 2010.

---

## Author Comment (AC2) · 25 Nov 2017

**Authors' Response to the Reviewer Comments**

**Manuscript Ref.: hess-2017-533**

**Title: Assessment of the Weather Research and Forecasting (WRF) Model for Extreme Rainfall Event Simulations in the Upper Ganga Basin**

**Authors:** Ila Chawla, Krishna K. Osuri, Pradeep P. Mujumdar, and Dev Niyogi

We sincerely thank the reviewers for their comments on the manuscript and offering their suggestions and critical input that has helped improve the manuscript. We provide here our replies to the reviewers' comments and highlight the changes made in the revised manuscript based on the comments. Sections which are modified in the revised manuscript are mentioned in this document.

Responses to the comments of Referee #2:

**General Comments:**

The paper provides results of a case study for an extreme event in northern India using WRF with multiple physics combinations. It evaluates rainfall primarily against observational data at stations and derived from TRMM. This focuses on the heavy rain period of 15-18 June 2013, and therefore conclusions about performance of the different simulations cannot be taken to be very robust in their usefulness for other cases. This limits the usefulness of this study. While it is interesting that certain physics combinations performed well and some parameterizations did well in several different combinations, it was not clearly presented.

Some physics schemes showed greater sensitivity to other schemes combined with C1 them, as seen in Fig. 10 for example. But showing the absolute difference hides some information that could have been seen from the difference itself that may have negative values.

It is also hard to determine signal from noise in Figures 7,8, and 11 when a station mean might have been of value in showing overall trends.

A major problem I have is with the use of CORDEX data. CORDEX is downscaled from climate model simulations and would therefore not be expected to bear any resemblance to real weather on any specific date. These are not comparable with weather models driven by real analysis boundary conditions. Sampling CORDEX on a particular date of a heavy rainfall event therefore will not be a fair comparison because it is more likely to miss such events entirely while it may have them on other days depending on which global data is used. It is no surprise that CORDEX

runs underestimate heavy events on a particular date, which does not mean they underestimate them in general. Such data can only be used qualitatively to see if they can capture heavy events over many years that they are run using frequency analyses. I therefore suggest that the CORDEX part serves no value for this case study, and if the authors are using CORDEX it can only be in the context of the climatology of heavy events and whether the observed peak can be captured at other times of those runs. Maybe these runs never have such events, which may be useful to know, or maybe they have them too frequently, also useful.

**Response:**

*"The paper provides results of a case study for an extreme event in northern India using WRF with multiple physics combinations. It evaluates rainfall primarily against observational data at stations and derived from TRMM. This focuses on the heavy rain period of 15-18 June 2013, and therefore conclusions about performance of the different simulations cannot be taken to be very robust in their usefulness for other cases. This limits the usefulness of this study. While it is interesting that certain physics combinations performed well and some parameterizations did well in several different combinations, it was not clearly presented."*

We thank the reviewer for the comment. There are two notable changes made in the revised manuscript that directly address the concerns raised. First, we have taken off the CORDEX aspects since the focus was getting diluted regarding the overall goal of the paper; and second, we have built on the study by not restricting on a single case and in fact adding five additional heavy to extremely heavy rainfall events. These cases correspond to the individual month of the monsoon season (June to September), that occurred in the upstream region of the UGB and are now included in the revised manuscript to further strengthen the conclusions and the choice of physics scheme.

**Action:** The document is thoroughly edited and requisite changes are made in the revised manuscript. We hope the discussion on the results is clear now.

*"Some physics schemes showed greater sensitivity to other schemes combined with them, as seen in Fig. 10 for example. But showing the absolute difference hides some information that could have been seen from the difference itself that may have negative values."*

**Response:** We agree with the reviewer that presenting the absolute difference may not bring out some information which may otherwise be important for understanding the effect of interactions between different parametrization schemes.

**Action:** Figure 10 is changed in the revised manuscript showing the difference in the simulated rainfall. Discussion pertaining to the figure is changed accordingly in the revised manuscript. The revised figure is presented here as Figure R.1 for reference.

[Figure]

**Figure R.1.** Difference in simulated rainfall due to PBL, CU and MP parametrization schemes corresponding to (i) Domain 1; (ii) Domain 2a; and (iii) Domain 2b over the UGB region.

*"It is also hard to determine signal from noise in Figures 7, 8, and 11 when a station mean might have been of value in showing overall trends."*

**Response:** We agree.

**Action:** Figure 7, 8 and 11 are changed in the revised manuscript. An additional column on the x-axis is added representing the spatial mean value for all the models. Discussion on the figures is accordingly changed in the revised manuscript. The revised figures are presented here as Figure R.2, R.3 and R.4 for reference.

[Figure]

**Figure R.2.** Root mean square error (top panel) and mean absolute error (bottom panel) computed temporally for (i) Domain 1; (ii) Domain 2a; and (iii) Domain 2b for (a) to (p)* WRF configurations.

*Refer to Appendix A (Table A.1) for the list of the WRF configurations.

[Figure]

**Figure R.3.** Scale error (*SE*) in (a) to (p)* WRF configurations for 18 locations in the UGB for (i) Domain 1; (ii) Domain 2a; and (iii) Domain 2b.

*Refer to Appendix A (Table A.1) for the list of the WRF configurations.

[Figure]

**Figure R.4.** Scatter plot (top panel) and mean difference in rainfall (bottom panel) for the observed rainfall data and the WRF simulations (for (b) and (p) configurations) pertaining to Noah and Slab LSMs corresponding to (i) Domain 1; (ii) Domain 2a; and (iii) Domain 2b.

*"A major problem I have is with the use of CORDEX data. CORDEX is downscaled from climate model simulations and would therefore not be expected to bear any resemblance to real weather on any specific date. These are not comparable with weather models driven by real analysis boundary conditions. Sampling CORDEX on a particular date of a heavy rainfall event therefore will not be a fair comparison because it is more likely to miss such events entirely while it may have them on other days depending on which global data is used. It is no surprise that CORDEX runs underestimate heavy events on a particular date, which does not mean they underestimate them in general. Such data can only be used qualitatively to see if they can capture heavy events over many years that they are run using frequency analyses. I therefore suggest that the CORDEX part serves no value for this case study, and if the authors are using CORDEX it can only be in the context of the climatology of heavy events and whether the observed peak can be captured at other times of those runs. Maybe these runs never have such events, which may be useful to know, or maybe they have them too frequently, also useful."*

**Response:** We agree.

**Action:** Analysis related to the CORDEX data is removed from the revised manuscript.

**Specific Comments:**

**Comment 1:** line 222. It said KF is shallow convection when it has both deep and shallow convection.

**Response:** This sentence has been clarified.

**Action:** Correction is done in the revised manuscript.

**Comment 2:** line 231. Both PLin and WSM6 are 6-class if vapor is included as a class.

**Response:** This is restated in the revised manuscript.

**Comment 3:** Figure 5. It is noted that domain 2b has no cumulus scheme within the domain, yet shows sensitivity to cumulus schemes, presumably through its boundaries and parent domain.

**Response:** That is correct. The effect of the cumulus scheme in Domain 2b is through the boundary feedback. It is generally accepted that at finer spatial resolution, such as 3 km or less, representing convective precipitation explicitly may yield better simulation results (Sikder and Hossain, 2016; Pieri et al., 2015; Yu and LEE, 2010; Done et al., 2004).

**Comment 4:** Figure 6. Over what period is this rainfall summed? Is it interpolated to the station point?

**Response:** Rainfall is summed over the 4 days' period (15 − 18 June 2013). Because of the complex variability in the domain, rainfall is not interpolated to the station location. The grid point closest to the gauge location is considered for comparison.

**Action:** Following line is added in the revised manuscript:

*"Figure 6 summarizes the comparison of WRF rainfall with rain gauge observations, accumulated over the 4 days' period (15 − 18 June 2013) for the three domains. For comparison, grid points from the WRF domains closest to the gauge location are considered."*

**Comment 5:** line 288. Presumably the complex terrain is also a factor in the bias at stations. Even at 3 km, there may be flow and rainfall differences because the model does not fully resolve all the terrain details.

**Response:** This is correct and is one of the challenge in simulating this complex region.

**Action:** Following line is added in the revised manuscript:

*"(iv) inability of the model to fully resolve the complex topography (Cardoso et al., 2013;Argüeso et al., 2011;Chevuturi et al., 2015)".*

**Comment 6:** Figure 7. Is this the MAE for the total 4-day precipitation at each station?

**Response:** No, *MAE* is computed for daily rainfall over the 4 days' period. The absolute error between the simulations and the observations is computed for each day and then averaged over the 4 days to get the mean absolute error.

**Action:** The sentence is modified in the revised manuscript and now reads as:

*"To assess the performance of the WRF simulations, quantitative scores (MAE and RMSE) with respect to the observed data are computed for daily rainfall data, which is then averaged over the 4 days. The results are shown in Figure 7."*

**Comment 7:** Care should be taken when suggesting Goddard is best especially as it has less overall precipitation. Lower precipitation itself may lead to lower absolute errors than schemes that more correct total amounts in the wrong places. Smoother precipitation fields may always score favorably in MAE. Total precip is an important factor to evaluate.

**Response:** We agree. We have added spatially averaged *MAE* and *RMSE* values in Figure 7, 8 and 11 in the revised manuscript. Through the spatially averaged values, it is noted that the errors are lowest when the configuration with MYJ PBL, BMJ CU, and Goddard MP is used.

Furthermore, 5 additional heavy to extremely heavy rainfall events, each corresponding to the individual month of the monsoon season (June to September), that occurred in the upstream region of the UGB are performed and included in the revised manuscript. These results also indicate that the configuration with MYJ PBL, BMJ CU, and Goddard MP is indeed the 'best' in simulating the spatial and temporal variability of the extremely heavy rainfall over the upstream region of the UGB.

**Comment 8:** line 316. BMJ even does better in 2b where it is not used and only would contribute through the boundaries, and this is surprising.

**Response:** The 'good' performance of BMJ CU is mentioned based on the overall results obtained for all the three domains. However, as mentioned in the response to Comment 3 (Referee #2), in Domain 2b, although cumulus scheme is not considered, still the simulations are sensitive to cumulus parameterizations used in the outer domain. This is typically due to the boundary conditions provided by the parent domain. This feature is also observed in several other studies (Sikder and Hossain, 2016; Pieri et al., 2015; Yu and LEE, 2010; Done et al., 2004), wherein it was seen that resolving convective precipitation explicitly at higher resolution gives better simulation results.

**Comment 9:** line 354. SE has not been defined. It looks like a ratio of model to observed variance.

**Response:** Scale Error (SE) is the ratio of standard deviation of model simulations to the observed standard deviation and is now explicitly defined.

**Action:** Following line is modified in the revised manuscript and it now reads as:

*"Ability of the WRF model configuration to simulate an extreme rainfall event is evaluated by comparing the simulated rainfall with the observations through indices such as Scale Error (SE), which is the ratio of standard deviation of model simulations to the observed standard deviation and Coefficient of Variation (CV) in addition to MAE, RMSE and β."*

**Comment 10:** Figure 9. It is confusing that colors are used both for rainfall and CV. Perhaps rainfall can be contoured.

**Response:** The figure is modified in the revised manuscript. Revised Figure 9 is presented here as Figure R.5.

[Figure]

**Figure R.5.** Coefficient of Variation (*CV*) value across different WRF configurations in the UGB for (i) Domain 1; (ii) Domain 2a; and (iii) Domain 2b.

**Comment 11:** line 416-424. Slab underestimates rainfall. This raises the issue of its moisture availability value in this region. How high is it? Can a higher value give a better rainfall?

**Response:** The Slab run uses the default soil moisture availability term, unlike the Noah model which has a prognostic soil moisture (and temperature) equation. In response to the reviewer's comment, we analyzed soil moisture values in Slab and Noah runs and highlight that a dry soil (soil moisture less than 0.05 $m^3/m^3$) persists through the integration in Slab runs. In the case of Noah, with the availability of rainfall induced soil moisture changes, soil moisture is found to vary between 0.25 $m^3/m^3$ to 0.45 $m^3/m^3$. Indeed, higher surface moisture leads to improved mass flux, by aiding convective updrafts and diabatic heating in the boundary layer that contributes to low level positive potential vorticity or convective potential which leads to enhanced rainfall (Osuri et al., 2017). In our prior study (Osuri et al., 2017), we studied the soil moisture and soil temperature impact on severe convection over India and demonstrated that drier the soil, lesser the rainfall and

vice versa. (Rajesh et al., 2016) also obtained improved rainfall prediction with the realistic soil conditions for Uttarakhand heavy rainfall case.

**Action:** Following lines are added in the revised manuscript:

*"Better performance of using the Noah model could be attributed to the temporal evolution of soil moisture fields. Analyzing the soil moisture in Slab and Noah model, the soil is noted to be relatively dry in Slab (soil moisture less than 0.05 $m^3/m^3$) and the value is constant throughout the model run (since in the Slab model there is no prognostic soil moisture term). In case of Noah, soil moisture varies in response to the rainfall and is found to vary between 0.25 $m^3/m^3$ to 0.27 $m^3/m^3$. Higher surface moisture conditions improve mass flux, convective updrafts and diabatic heating in the boundary layer that contributes to low level positive potential vorticity or convective potential which leads to enhanced rainfall potential (Osuri et al., 2017a). The importance of representing soil moisture variability over India for extreme weather conditions is also highlighted through this work."*

**Comment 12:** Major issue with using CORDEX as it is. See above comments.

**Response:** CORDEX section (Section 2.2 (old manuscript) and part of Section 3.2) is removed from the revised manuscript.

**Comment 13:** Major issue with conclusions being drawn from one case. See above.

**Response:** We have performed analysis for additional events to support our conclusion as stated above. The simulations for five different (additional) heavy to extremely heavy rainfall events, each corresponding to the individual month of the monsoon season (June to September), that occurred in the upstream region of the UGB are now included in the revised manuscript.

**Action:** Following details are added in the revised manuscript:

Section 2.1:

[revised manuscript text omitted]